# Potential Pathogenic Impact of Cow’s Milk Consumption and Bovine Milk-Derived Exosomal MicroRNAs in Diffuse Large B-Cell Lymphoma

**DOI:** 10.3390/ijms24076102

**Published:** 2023-03-23

**Authors:** Bodo C. Melnik, Rudolf Stadler, Ralf Weiskirchen, Claus Leitzmann, Gerd Schmitz

**Affiliations:** 1Department of Dermatology, Environmental Medicine and Health Theory, University of Osnabrück, D-49076 Osnabrück, Germany; 2University Clinic for Dermatology, Johannes Wesling Medical Centre, D-32429 Minden, Germany; 3Institute for Molecular Pathobiochemistry, Experimental Gene Therapy and Clinical Chemistry (IFMPEGKC), RWTH University Hospital Aachen, D-52074 Aachen, Germany; 4Institute of Nutrition, University of Giessen, D-35390 Giessen, Germany; 5Institute for Clinical Chemistry and Laboratory Medicine, University Hospital of Regensburg, University of Regensburg, D-93053 Regensburg, Germany

**Keywords:** B cell differentiation, B cell proliferation, BCL6, BLIMP1, diffuse large B-cell lymphoma, lymphomagenesis, microRNA, milk, milk-derived exosome

## Abstract

Epidemiological evidence supports an association between cow’s milk consumption and the risk of diffuse large B-cell lymphoma (DLBCL), the most common non-Hodgkin lymphoma worldwide. This narrative review intends to elucidate the potential impact of milk-related agents, predominantly milk-derived exosomes (MDEs) and their microRNAs (miRs) in lymphomagenesis. Upregulation of PI3K-AKT-mTORC1 signaling is a common feature of DLBCL. Increased expression of B cell lymphoma 6 (BCL6) and suppression of B lymphocyte-induced maturation protein 1 (BLIMP1)/PR domain-containing protein 1 (PRDM1) are crucial pathological deviations in DLBCL. Translational evidence indicates that during the breastfeeding period, human MDE miRs support B cell proliferation via epigenetic upregulation of BCL6 (via miR-148a-3p-mediated suppression of DNA methyltransferase 1 (*DNMT1*) and miR-155-5p/miR-29b-5p-mediated suppression of activation-induced cytidine deaminase (*AICDA*) and suppression of BLIMP1 (via MDE let-7-5p/miR-125b-5p-targeting of *PRDM1*). After weaning with the physiological termination of MDE miR signaling, the infant’s BCL6 expression and B cell proliferation declines, whereas BLIMP1-mediated B cell maturation for adequate own antibody production rises. Because human and bovine MDE miRs share identical nucleotide sequences, the consumption of pasteurized cow’s milk in adults with the continued transfer of bioactive bovine MDE miRs may de-differentiate B cells back to the neonatal “proliferation-dominated” B cell phenotype maintaining an increased BLC6/BLIMP1 ratio. Persistent milk-induced epigenetic dysregulation of BCL6 and BLIMP1 expression may thus represent a novel driving mechanism in B cell lymphomagenesis. Bovine MDEs and their miR cargo have to be considered potential pathogens that should be removed from the human food chain.

## 1. Introduction

Diffuse large B-cell lymphoma (DLBCL) is a highly heterogeneous lymphoid neoplasm with variations in gene expression profiles, caused by genetic and epigenetic alterations. DLBCL is the most common type of non-Hodgkin lymphoma (NHL) worldwide, representing approximately 30–40% of all cases in different geographic regions [1,2]. Patients most often present with a rapidly growing tumor mass in single or multiple, nodal or extranodal sites [1,2]. Primary cutaneous diffuse large B cell lymphoma, leg type is an aggressive lymphoma with an inferior prognosis [3,4,5]. Recent DLBCL prevalence was estimated to be between 63,000 and 143,000 cases in the US [6]. The total number of incident and prevalent cases of DLBCL is expected to increase between 2020 and 2025 in the US and Western Europe [7]. The most popular classification by gene expression profiling subdivided DLBCL into three groups according to the cell-of-origin: the germinal center B-cell like (GCB) subtype, the activated B-cell like (ABC) subtype, and about 10–15% of cases being unclassifiable [2]. Patients with the GCB subtype usually have a better prognosis than patients with the ABC subtype [2,8,9]. Molecular pathology is complex and GCB and ABC tumors have different mutation and translocation profiles [2,10,11,12]. Gene mutations in DLBCL disturb various signaling pathways including histone modification, cell growth, proliferation, metabolism, differentiation, apoptosis, survival, homing/migration, response to DNA damage, B-cell receptor (BCR), signaling, Toll-like receptor (TLR) signaling, angiogenesis, and immunoregulation [10,13,14,15,16,17,18,19].

Among the well-established genetic aberrations, epigenetic deviations emerged as further important drivers of DLBCL lymphomagenesis [20,21,22,23,24,25,26,27,28]. Notably, the B cell lymphoma 6 (BCL6) proto-oncogene encodes a transcriptional repressor, which is required for germinal center (GC) formation and lymphomagenesis. Constitutive expression of BCL6 leads to DLBCL through activation-induced cytidine deaminase (*AICDA*)-mediated chromosomal translocations and mutations [10,29]. Recently, Jiao et al. [28] showed that AICDA and DNA methyltransferase 1 (DNMT1) induce BCL6 promoter methylation. Either loss of AICDA or DNMT1 with the instability of the AICDA-DNMT1 complex on the BCL6 promoter result in BCL6 promoter demethylation leading to increased BCL6 expression in DLBCL [28].

A variety of intrinsic and extrinsic pathogenic factors have been related to DLBCL pathogenesis. Prior radiation treatment, obesity, and smoking are most highly associated with DLBCL as well as infections with human immunodeficiency virus (HIV), Epstein–Barr virus (EBV), and human herpes virus 8 (HHV8) [30]. Drugs like phenytoin, digoxin, and TNF antagonists are also associated with lymphomagenesis as well as organic chemicals, pesticides, bisphenol A, phenoxy-herbicides, glyphosate, wood preservatives, dust, hair dyes, solvents, and prior chemotherapy [31,32,33,34,35,36,37,38].

Among dietary factors, total animal protein intake, meat, dairy, and milk consumption have been associated with an increased risk of NHL [39,40,41,42,43], whereas vegetable- and fruit-based diets reduce the risk of NHL including DLBCL [44,45,46,47,48]. Three recent large epidemiological studies discussed in the next chapter in more detail identified an increased risk of DLBCL by total dairy and cow’s milk consumption [49,50,51]. Cow’s milk is a major dietary exposure in industrialized countries often persisting over a lifetime [52]. Milk is not a simple nutrient, but operates as an endocrine signaling system of mammalian evolution enhancing mTORC1 activation [53,54].

It is the aim of this review to elucidate potential synergism in signal transduction between identified signaling pathways in DLBCL and milk-induced signaling pathways explaining the mechanistic link between cow’s milk consumption and DLBCL pathogenesis.

## 2. Epidemiological Evidence for Milk Intake and DLBCL Risk

In 2016, Wang et al. [49] performed a meta-analysis of 16 relevant articles related to dairy/milk consumption and NHL risk published up to October 2015. The pooled relative risks (RRs) (95% Confidence Intervals (CIs) of NHL for the highest vs. lowest category of the consumption of total dairy product, milk, butter, cheese, ice cream, and yogurt were 1.20 (1.02–1.42), 1.41 (1.08–1.84), 1.31 (1.04–1.65), 1.14 (0.96–1.34), 1.57 (1.11–2.20), and 0.78 (0.54–1.12), respectively. In subgroup analyses, the positive association between total dairy product consumption and the risk of NHL was found among case-control studies (RR = 1.41, 95% CI: 1.17–1.70, *p* = 0.368) but not among cohort studies (RR = 1.02, 95% CI: 0.88–1.17, *p* = 0.988). The pooled RRs of NHL were 1.21 (95% CI: 1.01–1.46, *p* = 0.140) for milk consumption in studies conducted in North America, and 1.24 (CI: 1.09–1.40, *p* = 0.245) for cheese consumption in studies that adopted validated food frequency questionnaires. A further NHL subtype analysis identified statistically significant associations between the consumption of total dairy product (RR = 1.73, 95% CI: 1.22–2.45, *p* = 0.67) and milk (RR = 1.49, 95% CI: 1.08–2.06, *p* = 0.33) and the risk of DLBCL. The dose-response analysis suggested that the risk of NHL increased by 5% and 6% for each 200 g/day increment in total dairy product and milk consumption, respectively [49].

In contrast, a systematic review and meta-analysis published in 2019 by Sergentanis et al. [55] found no association between milk and dairy product consumption and the risk of NHL based on three included studies published between 1996–2011 [44,56,57]. All of those reported on milk consumption. The pooled analysis pointed to a null association with NHL risk (pooled RR = 0.99, 95% CI: 0.85–1.15, *p* = 0.461) [55]. Conversely, the exposome-wide analysis within the European Prospective Investigation into Cancer and Nutrition study (475,426 participants) found 2,402 incident cases of B-cell lymphoma [51]. Notably, dairy product consumption was positively associated with B-cell lymphoma and DLBCL risk [51]. The prospective China Kadoorie Biobank study recruited ~0.5 million adults from ten diverse (five urban, five rural) areas across China during 2004–2008. A dairy intake of 50 g/day in urban regions exhibited an HR for lymphoma of 1.09 (95% CI: 0.89–1.34, *p* = 0.11), and for rural regions HR = 1.45 (95% CI: 1.07–1.96, *p* = 0.55) [50]. Taken together, recent epidemiological evidence points to an increased risk for NHL and in particular, DLBCL related to the consumption of milk and dairy products (Table 1).

## 3. Potential Milk-Related Factors Promoting DLBCL

### 3.1. Insulin-Like Growth Factor 1 Signaling in DLBCL

In the context of carcinogenesis, the most important functions of the insulin-like growth factor (IGF) family involve the intensification of proliferation and inhibition of cell apoptosis and effects on cell transformation through the synthesis of several regulatory proteins. The IGF axis controls survival and promotes metastases. Interactions of IGF axis components may be of a direct or indirect nature. The direct effects are linked to the activation of the PI3K/AKT/mTORC1 signaling pathway, stimulated by IGF-1 and IGF-1 receptor (IGF1R) activation. Activation of this signaling pathway increases mitogenesis and cell cycle progression but protects against different apoptotic stresses [58,59]. Sustained proliferation and evading apoptosis are critical hallmarks of cancers [60].

Recent evidence indicates that mitogenic IGF-1/IGF1R signaling plays a role in the pathogenesis of DLBCL [61,62,63,64,65]. DLBCL cell lines together with primary tumor cells derived from lymph nodes in four DLBCL patients were treated with the cyclolignan picropodophyllin (PPP), a selective inhibitor of IGF1Rs. PPP dose-dependently inhibited proliferation/survival in all DLBCL cell lines and primary cell preparations [61]. Remarkably, lower expression levels of Klotho, a physiological inhibitor of IGF1Rs [63], were observed in DLBCL patients and cell lines [64]. Enforced expression of Klotho could significantly induce cell apoptosis and inhibited tumor growth in DLBCL. Upregulation of Klotho resulted in declined activation of IGF1R signaling, accompanied by decreased phosphorylation of its downstream targets, including AKT and ERK1/2. Moreover, in a xenograft model treated with either Klotho overexpression vector or recombinant human Klotho administration presented restrained tumor growth and lower Ki67 staining [64]. Recently, Zhou et al. [65] observed aberrant activation of Hippo-YAP signaling in DLBCL. Loss of Hippo-Yes-associated protein (YAP) attenuated proliferation and induced cell cycle arrest in DLBCL cells [65]. Moreover, the downregulation of IGF1R expression led to a remarkable decrease in YAP expression. In contrast, exposure to IGF-1 promoted YAP expression and reversed the inhibition of YAP expression induced by IGF1R inhibitors [65]. IGF-1/IGF1R/YAP signaling may thus represent a new pathway in DLBCL tumorigenesis. Agarwal et al. [66] recently provided evidence that elevated levels of human oncoprotein Smoothened (SMO), a Frizzled-class-G-protein-coupled receptor, show a strong correlation with elevated levels of IGF1Rs and reduced survival in DLBCL patients. As an integral component of raft microdomains, SMO plays a fundamental role in maintaining high levels of IGF1Rs in lymphoma cells as well as IGF1R-associated activation of AKT (protein kinase B). Silencing of SMO increases lysosomal degradation and favors a localization of IGF1R to late endosomal compartments instead of early endosomal compartments from which much of the receptors would normally recycle. In addition, the loss of SMO interferes with the lipid raft localization and retention of the remaining IGF1Rs and AKT, thereby disrupting the primary signaling context for IGF1R/AKT. This activity of SMO is independent of its canonical signaling and represents a novel and clinically relevant contribution to signaling by the highly oncogenic IGF1R/AKT signaling axis [66]. Taken together, IGF-1/IGF1R signaling is involved in DLBCL tumorigenesis.

### 3.2. Milk-Induced IGF-1- and Amino Acid-Mediated mTORC1 Signaling

Regular consumption of commercial cow’s milk enhances circulating IGF-1 levels in children and adults [67,68,69,70,71,72,73,74,75,76]. There are two mechanisms leading to milk-mediated elevations of circulatory IGF-1 levels of the milk recipient: (1) Uncertain proportions of bovine milk IGF-1, which shares an identical amino acid sequence with human IGF-1 [77], may be absorbed in the human intestine. (2) Milk components, especially milk protein-derived amino acids (tryptophan, arginine, and methionine) may induce the synthesis and secretion of pituitary GH and hepatic IGF-1 into the circulation [78,79,80,81].

#### 3.2.1. Milk-Derived Essential Branched-Chain Amino Acids

Milk proteins, especially whey proteins and caseins, are rich sources of essential branched-chain amino acids (BCAAs) that activate mTORC1 [54]. In all mammals except Neolithic humans, milk and milk protein exposure is restricted to the lactation period, whereas humans may be exposed to bovine milk proteins over their whole lifetime [52,82]. It is postulated that the type of protein in the diet influences directly the intrinsic capacity of the B lymphocytes to respond to an immunogenic stimulus [83]. The humoral immune response of mice fed with lactalbumin (20 g/100 g diet) was found to be higher than that of mice fed casein (20 g/100 g), soy protein (20 g/100 g) and wheat protein (20 g/100 g) diets, respectively [83]. To investigate the possible influence of dietary protein type on the supply of B lymphocytes, bone marrow lymphocyte production has been examined by a radioautographic assay of small lymphocyte renewal and an immunofluorescent stathmokinetic assay of pre-B cells and their proliferation [84]. The humoral response of all mice fed the lactalbumin-enriched diet was found to be higher than that of mice fed a casein diet or a control diet [84]. Cross and Gill [85] reported that bovine whey protein concentrate can modulate the proliferation of murine T and B cells in a dose-dependent manner. Among the major milk proteins, whey protein is the richest source of the essential BCAA leucine, which is the key amino acid activating mTORC1 via a RAG GTP-ase-dependent mechanism [86,87,88,89,90]. In contrast, reduced mTORC1 activity in B lymphocytes has been demonstrated at reduced extracellular amino acid and leucine concentrations [91]. The amount of leucine per g milk protein is a constant ratio for all mammals in the range of 10 g leucine/100 g milk protein [92]. Of all animal proteins, whey proteins contain the highest amount of leucine (14%) as compared to meat (8% leucine) [93]. Furthermore, in comparison to meat, whey proteins differ remarkably in their intestinal absorption kinetics, due to their fast intestinal hydrolysis increasing postprandial plasma amino acid levels [94,95,96,97].

#### 3.2.2. L-Type Neutral Amino Acid Transporter 1 in DLBCL

L-type neutral amino acid transporter 1 (LAT1) is a heterodimeric membrane transport protein involved in the uptake of neutral amino acids such as leucine, isoleucine, valine, phenylalanine, tyrosine, tryptophan, methionine, and histidine [98]. Remarkably, recent evidence indicates that high LAT1 expression is a poor prognostic factor in NHL [99]. The LAT1 expression level in samples of NHL patients ranged from 1.9 to 99.2%, with a median of 42.4% (SD ± 29.5%). For the most prevalent subtypes, DLBCL had the highest median LAT1 expression (80.1%) [99]. The median LAT1 expression levels of the GCB type of DLBCL and non-GCB type were 70.4% ± 22.4 and 83.3% ± 11.6, respectively [99]. Of note, LAT1 expression correlated with Ki-67 expression and thus the proliferation of DLBCL cells [99]. Thus, persistent consumption of milk and milk proteins may boost LAT1/leucine/mTORC1-driven DLBCL proliferation.

#### 3.2.3. Glutaminolysis in DLBCL

B-cell lymphomas use glutamine to power the tricarboxylic acid (TCA) cycle to generate energy and metabolic precursors [9,100]. Glutamine is metabolized through glutaminolysis to produce α-ketoglutarate. DLBCLs are dependent on mitochondrial lysine deacetylase sirtuin 3 (SIRT3) for proliferation, survival, self-renewal, and tumor growth in vivo regardless of disease subtype and genetics [101]. Importantly, overexpressed SIRT3 maintains DLBCLs’ metabolism by potentiating the TCA cycle through anaplerotic glutaminolysis [101]. In contrast, SIRT3 depletion impairs glutamine flux to the TCA cycle via glutamate dehydrogenase and reduces acetyl-CoA pools, which in turn induces autophagy and cell death [101]. Translational upregulation of activating transcription factor 4 (ATF4) is coupled with anaplerotic metabolism in DLBCLs due to nutrient deprivation caused by SIRT3 driving rapid flux of glutamine into the TCA cycle [102]. The active proliferation and high metabolic demand of DLBCL cells lead to a shortage of non-essential amino acids and results in translational activation of ATF4, which can transcribe target genes for the importation of extracellular nutrients to maintain the amino acid flux [102]. Moreover, the metabolic profile of DLBCL cells in the extracellular matrix is markedly different from cells in a suspension environment [103]. Recent evidence indicates that the synergistic consumption and assimilation of glutamine and pyruvate enables DLBCL proliferation in an extracellular environment-dependent manner [103].

Glutaminolysis is also important for the activation of mTORC1 [104,105]. mTORC1 is activated by glutamine and leucine via glutaminolysis-derived α-ketoglutarate upstream of RAG. This may provide an explanation for the glutamine addiction of cancer cells [105].

#### 3.2.4. Milk Proteins: A Rich Source of Glutamine

Casein, the major protein fraction of cow’s milk and the major protein component of cheese, contains 9 g glutamine/100 g casein [106,107]. Thus, milk and dairy products, especially cheese, provide a rich source of glutamine, which may fuel DLBCL, enhancing TCA-driven tumor cell proliferation.

### 3.3. Activation of mTORC1 in DLBCL

mTORC1 serves as a rheostat that shapes differentiation along the B lineage, the pre-immune repertoire, and antigen-driven selection of mature B cells [91]. Aberrant and persistent activation of the PI3K/AKT/mTORC1 signaling pathway plays an important role in controlling the proliferation and survival of tumor cells in various types of malignancies, including DLBCL [108]. Aberrant and persistent activation of mTORC1 is often observed in malignant B cells such as NHL. Distinct mechanisms drive mTORC1 activation in the three most-common NHL types, i.e., DLBCL, follicular lymphoma, and mantle cell lymphoma. Constitutive activation of the B-cell receptor (BCR), PI3K, and TLR pathways is a hallmark of ABC-DLBCL [109]. Activation of these pathways is attributed to mutations in their components (most frequently CD79A/B, CARD11, TNFAIP3, and MYD88) and leads to chronic activation of NF-κB signaling, proliferation, and survival [110]. Within this context, the CBM (CARD11-BCL10-MALT1) complex acts as a supramolecular organizing center to set the activation threshold and amplify BCR signaling to the IKK complex and NF-κB [110,111,112]. Notably, it has recently been demonstrated in ABC-DLBCL cells that the MYD88, TLR9, and BCR form a supercomplex that co-localizes with mTORC1 on endolysosomes, where it drives pro-survival NF-κB and mTORC1 signaling [113]. Inhibitors of BCR and mTORC1 signaling cooperatively decreased the formation and function of the MYD88/TLR9/BCR supercomplex, providing mechanistic insight into their synergistic toxicity for MyD88/TLR9/BCR^+^ DLBCL cells [113]. In fact, mTORC1 activity is required to drive the increased expression of eIF4B, which is a feature of DLBCL [114]. Of note, the initiation factor of translation eIF4E, a downstream effector of mTORC1, has oncogenic effects in vivo and cooperates with C-MYC in B-cell lymphomagenesis [115]. Multi-level inhibition of the PI3K/AKT/mTORC1 signaling pathway in DLBCL showed significant anti-tumor effects in DLBCL [116,117,118,119,120,121,122,123,124]. Notably, rituximab combined with rapamycin synergically downregulated the PI3K/AKT/mTORC1 signaling pathway [118].

#### 3.3.1. Milk-Induced Activation of mTORC1

Milk is an endocrine signaling system that promotes the activation of mTORC1 and mTORC1-dependent translation [53,54]. In comparison to a cow’s milk-free diet, young mice that had additional access to commercial cow’s milk exhibited increased expression of the mTORC1 downstream kinase pS6K1 in white adipose tissue and liver [125]. Remarkably, the mTORC1 response measured in mouse skeletal muscle following ingestion of high-quality plant-based and insect proteins was dampened compared to whey protein [126]. Bovine milk, a feeding and signaling system for postnatal anabolism and growth, activates IGF-1-driven activation of mTORC1 as well as BCAA- and glutamine-mediated activation of mTORC1. Persistent consumption of cow’s milk may thus augment mTORC1-mediated pathways augmenting mTORC1-driven DLBCL lymphomagenesis.

#### 3.3.2. B-Cell Receptor Activation in DLBCL

The BCR signaling pathway is a crucial pathway of B cells, both for their survival and for antigen-mediated activation, proliferation, and differentiation [127]. Its activation is also critical for the genesis of many lymphoma types. BCR-mediated lymphoma proliferation may be caused by activating BCR-pathway mutations and/or by active or tonic stimulation of the BCR [128,129,130]. A substantial fraction of DLBCLs are addicted to oncogenic BCR and PI3K/mTORC1 signaling caused by different stimuli and various genetic aberrations [131,132]. It has been suggested that stimulation of BCR by specific antigens including autoantigens and antigens of infectious origin plays a pathogenic role in DLBCL [129,130]. Targeting BCR and downstream PI3K/AKT/mTORC1 signaling is a recent therapeutic approach in DLBCL [113,131,132].

### 3.4. Milk Peptide-Induced B-Cell Receptor Activation

Bovine milk proteins, whey proteins and caseins, are immunogens and allergens, even when proteins are present at very low concentrations. There are both conformational and linear epitopes, widely spread all along the protein molecules [133,134]. Bovine milk-derived peptides encrypting possible bioactive and/or immunogenic molecules originating from caseins, β-lactoglobulin, and minor milk proteins have recently been detected in human plasma after intake of pasteurized cow’s milk [135]. Cow’s milk protein allergy is a common condition encountered in young children [136,137], but may rarely also occur in adults [138]. Bovine milk proteins have been implicated to function as triggers for autoimmune diseases, especially type 1 diabetes mellitus [139,140,141,142,143], which has recently been linked to a moderately increased risk of NHL [144]. Cow’s milk protein sensitivity is related to irritable bowel syndrome in patients with primary Sjögren’s syndrome [145], which was associated with a 6.5-fold increased risk of NHL including DLBCL [146].

B-cell epitopes as a screening instrument for persistent cow’s milk allergy have been identified [147]. Immunoglobulin E (IgE) and IgG binding epitopes on β- and κ-casein have been detected in cow’s milk allergic patients [148]. Antibodies raised against peptide fragments of bovine α-s1-casein cross-react with the intact protein only when the peptides contain both B and T cell determinants [149]. Notably, bovine α-s1-casein and its peptides 61–110 and 91–110 contain both T and B cell determinants on α-s1-casein and can elicit peptide-native protein cross-reactive antibodies. Antibodies raised against peptide fragments of bovine α-s1-casein cross-react with the native protein but recognize sites distinct from the determinants on the protein [150]. Naïve mouse splenocytes stimulated with α- or κ-casein showed a similar immunogenic potential of both casein fractions. However, mice immunized with α-casein exhibited higher serum levels of IgG and IgA compared with mice immunized with κ-casein [151]. B-cell linear epitope analysis of milk proteins using in-silico tools showed κ-casein, β-casein, α-lactalbumin, β-lactoglobulin, αs1-casein, and αs2-casein contain 28, 36, 29, 28, 28, and 33 epitopes, respectively. Therefore, β-casein and αs2-casein have more B-cell epitope capacity [152].

Various physicochemical parameters can have an impact on the allergenicity of animal proteins [153]. Ultraheat treatment (UHT) of cow’s milk (100 °C/30 s) significantly altered the immunogenicity of most of the potent protein stimulants, which mostly coincided with their levels of protein denaturation. Pasteurization (72 °C/15 s) caused the least protein denaturation but altered the immunogenicity of several protein stimulants notably, including heat-stable caseins and α-lactalbumin [154]. In addition, it has been shown that heat treatment reduced the allergenicity of β-lactoglobulin by inducing conformational changes and by increasing its susceptibility to enzymatic digestion, both of which disrupted B-cell epitopes. Heat treatment alone did not alter the allergenicity of α-casein [155]. It is thus conceivable that raw and pasteurized milk in comparison to UHT and fermented milk exert stronger stimulatory effects on BCR signaling, which contributes to mTORC1 activation.

### 3.5. Estrogen Receptor-β Signaling in DLBCL

Recent evidence indicates that estrogens play a role in lymphomagenesis. According to the American Cancer Society Cancer Prevention Study-II Nutrition Cohort, a positive association between current postmenopausal combined use of estrogen and progestin and DLBCL has been observed [156]. Remarkably, increased expression of estrogen receptor-β (ERβ) has been detected in DLBCL cells [157,158,159]. Nuclear ERβ1 expression analysis in primary DLBCLs by immunohistochemistry revealed ERβ1 expression in 89% of the cases and was an independent prognostic factor for adverse progression-free survival in rituximab-chemotherapy treated DLBCL [157]. For nodal lymphoma, high ERβ expression (≥25%) was associated with poorer event-free survival independent of the international prognostic index with the adjusted hazard ratio (HR) of 2.49 (95% CI: 1.03–6.00, *p* = 0.042 [158]. ERβ is expressed at significantly higher levels in DLBCL compared to normal B cells, and ERβ plays a role in the protection against apoptosis in DLBCL [159]. Targeting of the ERβ with the selective estrogen receptor modulator tamoxifen reduced cell viability in all tested DLBCL cell lines [159]. In addition, tamoxifen-treated breast cancer patients showed a 38% reduced risk for DLBCL compared to breast cancer patients who did not receive tamoxifen [159].

As with estrogen receptor-α (ERα), estrogenic compounds including estrone (E1), 17β-estradiol (E2), and estriol (E3) activate ERβ. Relative to ERα, ERβ binds E3 and ring B unsaturated estrogens with higher affinity, while the reverse is true of E2 and E1 [160,161,162,163,164]. Interestingly, in lung cancer tissues and A549 cells, estrogen upregulated the IGF1R signaling through ERβ [165,166].

### 3.6. Milk-Derived Estrogens

Milk produced from “persistently” pregnant cows—the current routine praxis of the dairy industry to increase commercial milk yield—enhances milk estrogen concentrations. In pregnant cows, the predominant estrogen is E1 sulfate, which passes into milk [167]. Heat treatment (70 °C and 95 °C) does not affect E1 and E2 concentrations compared to unprocessed raw milk [168]. The concentration of E1 sulfate increases from 30 pg/mL in non-pregnant cows up to 151 pg/mL in pregnant cows at 40–60 days of gestation, and to a maximum level of 1000 pg/mL in cows at 220 days of gestation [169]. Farlow et al. [170] compared the estrogen concentrations of commercial cow’s and goat’s milk. They reported unconjugated and conjugated levels of E1 in regular whole cow’s milk of 129.9 ± 18.48 pg/mL and E2 levels of 28.19 ± 5.26 pg/mL, whereas in whole goat’s milk E1 and E2 levels were 42.48 ± 4.28 pg/mL and 17.87 ± 2.8 pg/mL, respectively [170]. The average milk concentrations of E1 (159 ng/kg) and E2 (6 ng/kg) in Swiss Holstein cows are comparable [171].

Pape-Zambito et al. analyzed E1 and E2 concentrations in pasteurized-homogenized milk and commercial dairy products [172]. E1 concentrations averaged 2.9, 4.2, 5.7, 7.9, 20.4, 54.1 pg/mL, and 118.9 pg/g in skimmed, 1%, 2%, and whole milks, half-and-half, cream, and butter samples, respectively. E2 concentrations averaged 0.4, 0.6, 0.9, 1.1, 1.9, 6.0 pg/mL, and 15.8 pg/g in skimmed, 1%, 2%, whole milks, half-and-half, cream, and butter samples, respectively [172]. The mean E2 mass in 237 mL of raw whole Holstein milk was 330 pg [173]. As the milk of pregnant dairy cows is pooled, commercial cow’s milk contains higher estrogen amounts compared to former times, when lactation of cows was synchronized and cows gave birth only in spring time. Maruyama et al. [169] analyzed the exposure to exogenous estrogen through the intake of commercial milk produced from pregnant cows in children and adults. Urine concentrations of E1, E2, E3, and pregnanediol significantly increased in all adults and children after intake of 600 mL/m^2^ of commercial cow’s milk. In prepubertal children, urinary excretion volumes of estrogens and pregnanediol significantly increased within 1–3 h. The net increase in E2 excretion from the basal E2 levels in urine (before the intake) was 39–109 ng/4 h in this study. These data indicate that the intake of estrogens from cow’s milk corresponds to the daily estrogen production rate in prepubertal boys [169]. Of note, not only milk but all dairy products, especially milk fat, contains estrogens that enhance the dietary estrogen exposure derived from cow’s milk and dairy products [174,175]. Milk products supply about 60–80% of ingested female sex steroids [174]. Taken together, it is conceivable that milk/dairy-derived estrogens via ERβ—IGF1R signaling may promote DLBCL cell proliferation.

#### 3.6.1. Bisphenol A in Lymphomagenesis

Bisphenol A (BPA) is a common chemical used in the manufacture of materials in polycarbonate and epoxy plastic products and can interfere with the immune system. BPA is a proven endocrine disruptor capable of mimicking or blocking ERs and altering hormone concentrations and metabolism [176,177,178]. There is increasing concern that BPA perturbs the immune system causing adverse health effects including cancer [179,180,181]. It has recently been shown that BPA-induced DNA damages promote lymphoma progression in human lymphoblastoid cells through the aberrant catenin-β1 signaling pathway [37]. Gene-network analysis of microarray data sets in human lymphoma tissues as well as in human cells with BPA exposure to explore module genes identified potential pathways for lymphomagenesis in response to BPA [37]. BPA exposure resulted in a disrupted cell cycle and DNA damage by activating catenin-β1 encoded on the catenin-β1 gene (*CTNNB1*), the initiator of the aberrant constructed CTNNB1-NFKB1-AR-IGF-1-TWIST1 pathway, which may potentially lead to lymphomagenesis [37]. Activated catenin-β1 suppresses the DNA-repair-associated genes *TP53* and *CDKN1A* [37]. Notably, dose-dependent colony formations of human lymphoblastoid TK6 cells visibly promoted by BPA treatment were significantly suppressed in *siCTNNB1*-transfected TK6 cells. In accordance, DLBCL showed a higher expression of β-catenin in contrast to reactive hyperplasia of lymph node tissues at both the mRNA and protein levels [182]. β-catenin is a key downstream effector of the Wnt/β-catenin pathway inducing the aging of mesenchymal stem cells through DNA damage response and the p53/p21 pathway [183]. Notably, constitutive activation of the DNA damage response pathway has been regarded as a novel therapeutic target in DLBCL [184]. In addition, increased TWIST expression, which is upregulated by IGF-1/IGF1R signaling [185], has also been observed in DLBCL [186]. Furthermore, canonical NF-κB signaling has been reported in DLBCL of the ABC type [187,188,189] and BPA is a known activator of NF-κB [190,191]. Of note, BPA also increases the expression of ERβ [192,193,194]. In thyroid tumor cells, BPA and E2 enhanced the expression of ERα/ERβ and GPR30 and activated AKT and mTORC1 [195].

Importantly, plasticizers like BPA modify the expression of oncogenic microRNAs (miRs) [196,197]. Notably, BPA exposure increased the expression of miR-21-5p in the majority of cell lines studied [196,197,198,199,200].

#### 3.6.2. Contamination of Commercial Milk with Bisphenol A

BPA enters into the milk chain via multiple pathways at various points during milk production including PVC tubing used during the milking process, transfer from bulk milk to storage tanks, and during milk processing and packing [201,202]. Santonicola and colleagues reported mean BPA concentrations were 0.757 µg/L in manually milked samples, 0.580 µg/L in mechanically milked samples, and 0.797 µg/L in milk from the cooling tank [203]. Quantifiable levels were detected in samples obtained from the raw milk storage tank, pasteurized milk from the storage tank, and packaged milk. The highest BPA contamination levels were detected in raw milk from the storage tank (mean 0.265 µg/L) [204]. In milk samples (supplied in plastic bottles) of the winter season, BPA levels were 0.17–0.32 mg/kg, whereas, in milk samples of the summer season, BPA levels of 0.77–1.59 mg/kg have been detected [205]. BPA occurred in the milk chain as a result of different stages of milking and reached the highest levels at the end of the milk chain. Although the dietary intake of BPA is below the European Food Safety Authority’s temporary tolerable daily intake, exposure to BPA through milk consumption may have a critical impact on lymphomagenesis as BPA augments β1-catenin/ERβ/IGF-1/IGF1R-signaling involved in the pathogenesis of DLBCL (Figure 1).

### 3.7. Viral Agents in DLBCL

Viral agents are regarded as potential drivers of DLBCL tumorigenesis [206,207]. Various reports suggest an association of viral infections with DLBCL, including Epstein–Barr virus (EBV) also known as human herpes virus 4 (HHV4) [208,209,210,211,212,213,214,215,216], human immunodeficiency virus HIV (HIV) [217,218,219,220,221], hepatitis B virus (HBV) [222,223,224,225,226], hepatitis C virus (HCV) [227,228,229], human T-cell leukemia virus type 1 (HTLV-1) [230], and Simian virus 40 (SV40) infections [231,232,233], while human herpes virus 8 (HHV8) also known as Kaposi sarcoma-associated herpesvirus (KSHV) [30,234,235,236] and human papilloma virus (HPV) [237,238] associations with DLBCL are only rarely observed. EBV-positive DLBCL, not otherwise specified (NOS), is an EBV-positive clonal B cell lymphoid proliferation [239], that plays a major role in all virus-associated DLBCLs [214].

The oncogenic miR-155-5p is the most frequently upregulated miR in EBV-positive B cell malignancies. Of note, miR-155-5p plays a key role in B-cell immortalization by EBV [240]. EBV nuclear antigen 2 (EBNA2) and the B cell transcription factor interferon regulatory factor 4 (IRF4) are known to activate transcription of the host cell gene from which miR-155 is processed (miR-155HG; BIC). EBNA2 also activates IRF4 transcription, indicating that EBV may upregulate miR-155 through direct and indirect mechanisms [241]. Remarkably, miR-K12-11 encoded by KSHV shows significant homology to cellular miR-155, including the entire miR ‘seed’ region. Evidently, viral miR-K12-11 functions as an orthologue of miR-155 and probably evolved to exploit a pre-existing gene regulatory pathway in B cells. Moreover, the known etiological role of miR-155 in B cell transformation suggests that miR-K12-11 may contribute to the induction of KSHV-positive B cell tumors in infected patients [242].

High levels of B cell activation are induced by miR-21-5p in circulating B cells and are seen in HIV-infected individuals compared to non-infected controls. Notably, miR-21 is overexpressed in activated B cells of HIV-infected patients, suggesting its assistance in maintaining B cell hyperactivation contributing to lymphomagenesis [243,244]. Oncogenic viruses, including EBV, HBV, HCV, and HPV, co-evolve with their hosts and cause persistent infections. The upregulation of host miR-21 manipulates key cellular pathways to evade host immune responses and then promote viral replication [245]. Exosomes released by oncogenic virus-infected cells play a key role in promoting or inhibiting cancer formation [246].

### 3.8. Bovine Meat and Milk Factors

In recent years, a variety of circular replicase-encoding single-stranded (CRESS) DNA viruses and unclassified virus-like DNA elements have been discovered in a broad range of animal species and environmental samples. These sequences have been assigned to the virus phylum *Cressdnaviricota*, where CRESS stands for “Circular Rep-Encoding Single-Stranded”, to describe the functional features of these isolates, and viricota is the suffix for phylum taxa [247]. Especially, DNA elements termed bovine meat and milk factors (BMMF) are suspected to act as co-factors in the development of colon and breast cancer. BMMFs represent small single-stranded circular DNA, predominantly isolated from sera, milk, and dairy products of Eurasian cattle [248,249] and subsequently identified in periglandular cells of colon and breast cancers [250]. BMMF Rep protein has been detected in close vicinity of CD68^+^ macrophages in the interstitial lamina propria adjacent to colorectal cancer tissues, suggesting the presence of local chronic inflammation [251]. Recently, Nikitina et al. [252] found pleomorphic vesicles, regularly identified by staining peritumor tissues of colorectal, lung, and pancreatic cancer for expression of BMMF Rep. This subgroup of BMMF1 proteins is involved in the replication of small single-stranded circular plasmids of BMMF, but most likely also contributes to the formation of pleomorphic vesicular structures found in the periphery of colorectal, lung and pancreatic cancers [252]. These infectious agents share characteristics of both bacterial plasmids and known viruses [253]. Recently, circular single-stranded DNA genomes have been identified in the milk of sheep and goats [254] as well as the milk of water buffaloes [255]. Thus, dairy cows, sheep, goats, and water buffaloes add to the dispersal of CRESS viruses and circular ssDNA elements, which enter the human food chain via milk consumption [248,249,254,255]. According to de Villiers and Zur Hausen, future transcription analyses of acute lymphatic leukemias searching for BMMF-like infections will be of substantial interest [256]. These surveys should also include DLBCLs, the most common NHL linked to milk consumption (Table 2).

### 3.9. Bovine Leukemia Virus

Bovine leukemia virus (BLV) is the causative agent of leukemia/lymphoma in cattle. Cattle are the natural host of BLV, which integrates into B cells, producing a lifelong infection. BLV is a deltaretrovirus closely related to HTLV-1 and HTLV-2 [257] and causes enzootic bovine leukosis, which is the most important neoplastic disease in cattle [258]. Infectious BLV is present in the colostrum and milk of most BLV-positive cows [259,260]. Recent evidence points to a genetic flux between cattle and humans [261]. Olaya-Galán et al. [262] detected BLV DNA in raw beef and fresh milk for human consumption reminiscent of findings on BMMF distribution. Notably, BLV DNA was detected in the buffy coat cells of the blood of human subjects [263]. The most likely route of BLV transmission to humans is a foodborne infection [263]. After more than two decades, the opinion that “BLV does not infect humans” has started to change. BLV has been identified in human breast cancers giving rise to the hypothesis that it could be one of the causative agents of this condition [264,265]. Remarkably, one BLV miR, BLV-miR-B4, shares partial sequence identity and shares common targets with the host miR-29 [266]. Overexpression of miR-29 is associated with B cell neoplasms suggesting a possible contribution to BLV-induced tumorigenesis [266]. Bovine milk exosome-derived (MDE) miR-29 combined with BLV-derived miR-B4 may thus enhance oncogenic miR-29 signaling.

## 4. Exosomal MicroRNAs in the Pathogenesis of DLBCL

B cell development is a very orchestrated pathway that involves several molecules, such as transcription factors, cytokines, and miRs, respectively. All these components maintain the ideal microenvironment to control B cell differentiation. MiRs are small non-coding RNAs that target mRNAs to control gene expression. These molecules could circulate in the body in a free form, protein-bounded, or encapsulated into extracellular vesicles (EVs) including exosomes [267]. EVs represent a heterogeneous group of cell-derived membranous structures comprising exosomes and microvesicles, which originate from the endosomal system or which are shed from the plasma membrane, respectively. They are present in biological fluids and are involved in multiple physiological and pathological processes, especially cell-to-cell communication. Exosomes are small, nano-sized (50–100 nm) EVs secreted by cells and carry nucleic acids, proteins, lipids, and other bioactive substances that play a role in the body’s physiological and pathological processes. They are secreted by all cells and circulate in all body fluids including milk. Exosomes are key mediators of several processes in cancer that mediate tumor progression and metastasis. These nano-vesicles, when secreted from cancer cells, are enriched in non-coding RNAs (e.g., miRs) complexed with the RNA-induced silencing complex (RISC), that mediate an efficient and rapid silencing of mRNAs at the recipient cell, reprogramming their transcriptome [268]. The recently updated “hallmark of cancer model” outlined by Hanahan and Weinberg [269] considers the roles of miRs in non-mutational epigenetic reprogramming promoting cancer development and progression [269]. There is current interest in the role of miRs and their relevant targets in B cell development, B cell activation, and B cell malignant transformation [270]. B cell-specific miR expression plays a key role in the regulation of GC responses and B cell neoplasia [271,272]. The biological role of EVs, especially exosomes and their miR cargo in DLBCL initiation and progression is a recent focus of DLBCL research [273]. Rutherford et al. [274] showed that DLBCLs secrete large quantities of CD63-, ALIX-, TSG101-, and CD81-positive exosomes. Importantly, DLBCL cells take up exosomes and their RNAs [274]. This observation highlights the importance of EVs and exosomes promoting aberrant biological programming of recipient cells, including pre-metastatic niche formation and tumor progression [274,275], also demonstrated in other malignancies such as B-cell chronic lymphocytic leukemia [276], melanoma [277,278], pancreatic cancer [279], hepatocellular carcinoma [280,281], breast cancer [282,283], and prostate cancer [284,285]. Of note, CLL-derived microvesicles (MV) can activate the AKT/mTORC1/p70S6K/hypoxia-inducible factor-1α axis in CLL-bone marrow stromal cells (BMSCs) with the production of vascular endothelial growth factor, a survival factor for CLL B-cells. Moreover, MV-mediated AKT activation led to modulation of the β-catenin pathway and increased expression of cyclin D1 and C-MYC in BMSCs [275]. EVs and exosomes and their miR may play a key role in the pathogenesis of DLBCL, particularly with regard to the exchange of genomic information [273,274]. In fact, it has been observed that normal B cells internalize DLBCL-derived exosomes resulting in miR expression differences observed in normal B cells that are specific to lymphoma-subtypes [286].

Increased levels of exosomal miR-125b-5p and miR-99a-5p in DLBCL patients’ serum were associated with shorter progression-free survival time, and they can predict chemotherapeutic efficacy. Remarkably, DLBCL exosome-derived miR-125b-5p and miR-99a-5p were significantly upregulated and present predictive biomarkers for DLBCL chemotherapy resistance [287]. Exosomal miR-125b-5p is also described as a potential prognostic predictor of chemoresistance in the serum of patients with DLBCL [287]. Notably, exosomes carrying miR-125b-5p can reduce DLBCL sensitivity to rituximab by inhibiting the expression of tumor necrosis factor necrosis factor-α induced protein 3 (TNFAIP3), also known as A20 [288]. TNFAIP3 is a negative regulator of NF-κB that has been implicated as a tumor suppressor in multiple types of B-cell lymphoma including DLBCL [13,288,289,290,291,292]. A large percentage of DLBCL cases (55%) that have an MYD88 mutation also harbor a loss of TNFAIP3 [293].

B lymphocyte-induced maturation protein 1 (BLIMP1) also known as PR domain-containing protein 1 (PRDM1) plays a pivotal role in gene regulation required for B cell function and terminal differentiation and plasmablast formation [294,295]. BLIMP1 represses BCL6, a GC-restricted transcriptional repressor for GC formation [296,297,298,299]. BCL6 is a transcriptional repressor often expressed constitutively in DLBCL due to mutations of its genomic locus. BCL6 mediates aberrant survival, proliferation, genomic instability, and differentiation blockade in DLBCL cells [300,301,302]. In addition, BCL6 suppresses the expression of p53 and modulates DNA damage-induced apoptotic responses in GC B-cells. Of note, BCL6 represses p53 transcription by binding two specific DNA sites within the p53 promoter region and, accordingly, p53 expression is absent in GC B-cells where BCL6 is highly expressed [303]. Importantly, BCL6 overexpression represses BLIMP1, suppressing plasma cell differentiation [304].

The majority of *TP53* mutations in human DLBCL are accompanied by loss of p53 function [305]. Pascual et al. [306] recently demonstrated that DNA damage response by p53 is a central mechanism suppressing the pathogenic cooperation of IKK2ca-enforced canonical NF-κB and impaired differentiation resulting from BLIMP1 loss in ABC-DLBCL lymphomagenesis. Conditional deletion of p53 in mouse GCBs strongly synergized with IKK2 activation and BLIMP1 loss to promote GC-derived lymphomagenesis. Thus, there is a close interaction between BLIMP1, BCL6, and p53 in the pathogenesis of DLBCL. BLIMP1 has been substantiated as a key tumor suppressor of DLBCL [307,308,309,310,311,312,313,314,315]. Mandelbaum et al. [313] and Calado et al. [314] unequivocally demonstrated that BLIMP1 functions as a tumor suppressor and guardian of ABC-like DLBCL lymphomagenesis. Loss of BLIMP1 function contributes to the overall poor prognosis of ABC-DLBCL patients [316,317]. Furthermore, dysregulated multidrug resistance protein 1 (MDR1; ABCB1) by BLIMP1 is involved in the doxorubicin resistance of non-GC B-cell-like DLBCL [318].

Notably, loss of BLIMP1 was associated with MYC overexpression and decreased expression of p53 pathway molecules [317]. BLIMP1 represses C-MYC promoter activity in a binding site-dependent manner [319]. The repression of C-MYC by BLIMP1 is a critical step for terminal B cell differentiation [319]. The upregulation of C-MYC gene expression induced by *PRDM1* inactivation apparently plays a crucial role in the development of DLBCL [320].

### 4.1. MicroRNA-Mediated Transcriptional Regulation in DLBCL

BLIMP1 has been identified as a key immune gene hub regulated by miRs [321]. Inactivation of BLIMP1 not only results from *PRDM1* mutations but also from miR-mediated downregulation of BLIMP1 expression. Ma et al. [322] demonstrated that BLIMP1 suppression via EBV-miR-BHRF1-2 plays a potential role in EBV-induced lymphomagenesis. Recently evidence indicates that certain miRs act as oncogenes in DLBCL pathology [323]. In the following section pivotal DLBCL-related miRs that are also signature miRs of cow’s milk and milk exosomes will be discussed in more detail.

#### 4.1.1. MicroRNA-let-7 Over-Expression in DLBCL

High levels of let-7a and miR-9 in HL cell lines correlated with low levels of BLIMP1 [324]. Let-7, in particular let-7b, is overexpressed in DLBCL relative to normal GC B-cells, suggesting that this miR is deregulated. Thus, abnormal epigenetic downregulation of BLIMP1 by let-7 and other miRs may represent an alternative epigenetic mechanism reducing BLIMP1 function in a subset of DLBCL [325].

#### 4.1.2. MicroRNA-125b Over-Expression in DLBCL

Malumbres et al. [326] provided experimental evidence that GC-enriched hsa-miR-125b down-regulates the expression of IRF4 and BLIMP1. Dysfunction of *TP53*, including miR regulations, copy number alterations of the p53 pathway and p53 itself, dysregulation of p53 regulators, and somatic mutations by abnormal p53 function modes, play an important role in DLBCL generation, progression, and invasion [327]. Importantly, miR-125b-5p is a key negative regulator of p53 expression and targets the 3′ untranslated region (3′ UTR) of *TP53* mRNA [328,329]. Li et al. [330] reported mutations in the 3′ UTR of *TP53* mRNA that modify miR-125b binding and the expression of p53 affected the response to therapy in DLBCL patients. Furthermore, miR-125b-5p targets MAX dimerization protein 4 (MXD4) thereby enhancing C-MYC activity [331].

#### 4.1.3. MicroRNA-21 Over-Expression in DLBCL

Elevated levels of miR-21-5p have been found in the serum of patients with DLBCL [332,333,334,335,336,337,338]. miR-21-5p contributes to characteristic miR signatures in DLBCL [287,339,340,341]. High miR-21-5p expression is associated with the ABC subtype of DLBCL [340]. miR-21-5p is the most frequently deregulated miR in malignancy, including B-cell lymphomas, and it has oncogenic potential downstream of STAT3 [342]. Remarkably, BLIMP1 binds to the promoter of pri-miR-21 and represses pri-miR-21 expression [342]. Using in situ hybridization, miR-21 expression was also detected in the stromal compartment of 73.2% of DLBCL [343]. DLBCL patients with higher miR-21 expression have shown significantly worse overall survival than those with lower miR-21 expression [335]. Inada et al. [334] confirmed that miR-21-5p is not only increased in serum but also in exosome-enriched serum of patients with DLBCL. Circulating exosomal miRs can be useful noninvasive biomarkers for the diagnosis of DLBCL and for improving the identification of patients with poor outcomes [344].

Whereas C-MYC induces the expression of miR-21, miR-21 in turn targets MAX dimerization protein 1 (*MXD1*) mRNA and downregulates MXD1 protein. The resultant decrease in MXD1 promotes the formation of a C-MYC-MAX heterodimer, leading to sustained C-MYC activation [345]. Thus, C-MYC/miR-21/MXD1 represents a positive-feedback loop that is critical for the maintenance of B-cell lymphoma survival. Targeting miR-21 with the bendamustine derivative NL101 blocked the C-MYC/miR-21/MXD1 loop, a potential therapeutic strategy of C-MYC-directed lymphoma therapy [345]. Von Hippel-Lindau (*VHL*) mRNA is another direct target of miR-21-5p in DLBCL [346]. In contrast, curcumin is a known suppressor of miR-21 [347] and decreases miR-21 levels through both increasing miR-21 exosome exclusion from the cells and inhibiting the transcription of the miR-21 gene (*MIR21*) in the cells by binding to its promoter [347]. In fact, curcumin exerted its anti-proliferation, anti-migration, anti-invasion, and pro-apoptosis functions, at least partly, by repressing miR-21 and regulating VHL expression in the DLBCL cell line [346]. Furthermore, miR-21-5p inhibits phosphatase and tensin homolog (PTEN) and programmed cell death protein 4 (PDCD4), thus increasing PI3K/AKT-mediated cell proliferation and cell survival as well as eIF4A and eIF4G-mediated translation [347]. In addition, miR-21 via activation of AKT increases NF-κB signaling [347,348,349]. miR-21 level was inversely correlated with the levels of FOXO1 and PTEN in DLBCL cell lines [350,351]. Reporter-gene assay showed that miR-21 directly targeted and suppressed FOXO1 expression, and subsequently inhibited BCL2-like 11 (BIM) transcription in DLBCL cells. miR-21 also down-regulates PTEN expression and consequently activates the PI3K/AKT/mTORC1 pathway, which further decreases FOXO1 expression. In contrast, FOXO1 activation is a required effector for spleen tyrosine kinase (SYK) and AKT inhibition in tonic BCR signal-dependent DLBCL [352]. Moreover, miR-21 inhibitor suppressed the expression and activity of multidrug resistance protein 1 (MDR1), thereby sensitizing DLBCL cells to doxorubicin [350]. In fact, miR-21 exhibited significantly higher plasma levels in patients with DLBCL unresponsiveness to treatment [353], whereas inhibition of miR-21 induced suppression of proliferation and invasion, as well as increased apoptosis in DLBCL [354]. According to a recent meta-analysis, poor overall survival has been associated with high expression of miR-21 in tumor tissue of DLBCL [355]. Recently, a novel prognostic model based on four circulating miRs (miR-21, miR-130b, miR-155, and miR-28) in DLBCL with implications for their roles of myeloid-derived suppressor cells (MDSC) and Th17 cells in lymphoma progression [356]. Remarkably, inhibition of miR-21 reduced the expression of IGF-1 [356]. The high-risk group of the 4-circulating miRs prognostic model was characterized by activation of RAS protein signal transduction. Both IGF-1 and JUN were two key regulators of the RAS cascade, enhancing lymphoma development [356]. As a mechanism of action, IGF-1 and JUN are positively regulated by miR-21 [356]. There is recent interested in miR-based therapies in B-cell NHL [357].

#### 4.1.4. Over-Expressed MicroRNA-155 in DLBCL

The evolutionarily conserved miR-155 plays an important role in the immune system, specifically in regulating GC reaction to produce an optimal T cell-dependent antibody response [358]. miR-155-5p and miR-155-3p are processed from the B-cell integration cluster (BIC) gene (now designated, *MIR155* host gene or *MIR155HG*). miR-155-5p is highly expressed in both activated B and T cells and monocytes/macrophages and proliferating lymphoblastoid cell lines [240,359]. miR-155 can be processed from sequences present in *BIC* RNA, a spliced and polyadenylated but non-protein-coding RNA that accumulates in lymphoma cells [360]. Isolates of several types of B-cell lymphomas, including DLBCL have 10- to 30-fold higher copy numbers of miR-155 than do normal circulating B cells [360]. Significantly higher levels of miR-155 are present in DLBCLs with an ABC phenotype than with the GC phenotype [360]. In fact, oncogenic miR-155 is highly expressed in the non-GC B-cell or activated B-cell subtype of ABC-DLBCL [361]. In accordance, miR-155 expression levels in formalin-fixed/paraffin-embedded tissue of patients with DLBCL were significantly higher in de novo DLBCL patients compared to controls [362]. PI3K is activated in DLBCL cells due to chronic or tonic BCR signaling [363]. miR-155 is potentially involved in the upregulation of BCR-PI3K-AKT-mTORC1 signaling DLBCL. Huang et al. [364] reported a link between miR-155 overexpression in DLBCL and overactivated PI3K-AKT signaling. Notably, the PI3K regulatory subunit 1 (*PIK3R1*; also known as p85α), which is a negative regulator of the PI3K-AKT pathway, is a direct target of miR-155-5p and miR-21-5p [364]. Knockdown of miR-155 in OCI-Ly3 cells diminished AKT activity [364]. PTEN has been identified as a further target of miR-155 [365,366]. It has recently been reported that miR-155 represses the mTOR phosphatase DEP domain-containing protein 6 (*DEPD6*) also known as DEP domain-containing mTOR-interacting protein (DEPTOR) as well as C-CBL (SYK ubiquitin E3 ligase) [367]. DEPTOR protein expression was markedly lower in more aggressive non-GCB DLBCLs than in GCB tumors [367]. DEPTOR is capable of inhibiting the kinase activity of mTOR of both mTORC1 and mTORC2 [368,369]. AKT Thr308 being phosphorylated by phosphoinositide-dependent kinase 1 (PDK1), the plasma membrane-localized and activated mTORC2 further phosphorylates AKT at Ser473 [370]. Thus, miR-155 upregulates PI3K-AKT-mTORC1 signaling at multiple inhibitory checkpoints. Upregulated miR-155, mTORC1, pS6K1, and downregulated DEPTOR were associated with the treatment-resistant group in gastric DLBCL [371]. The inositol polyphosphate-5-phosphatase (*INPP5D* also known as SHIP1) is another target of miR-155-5p [372]. SHIP dephosphorylates the PI3K product and lipid second messenger phosphatidylinositol-3,4,5-trisphosphate [PI(3,4,5)P3] to produce phosphatidylinositol-3,4-bisphosphate [PI(3,4)P2] [373,374]. SHIP plays a critical role in the termination of PI [3,4,5]P(3) signals that follow BCR aggregation [374]. B-cell precursors from SHIP-deficient mice progress more rapidly through the immature and transitional developmental stages and SHIP-deficient B cells have increased resistance to BCR-mediated cell death [374]. Diminished SHIP1 expression in DLBCL resulted in autocrine stimulation by tumor necrosis factor-α (TNFα) [375]. Whereas miR-155 downregulates SHIP expression increasing PI3K-AKT signaling, a novel SHIP1 activator, AQX-435, reduced AKT phosphorylation and growth of DLBCL in vivo and cooperated with ibrutinib for tumor growth inhibition [376]. DLBCL cases with elevated circulating levels of miR-155 had shorter overall survival than those with a lower miR-155 expression [377]. Notably, a significant increase in plasma exosomal miR-155, let-7g, and let-7i levels and exosome concentration in refractory/relapsed patients compared to responsive patients and patients receiving R-CHOP has been reported [378]. In contrast, the inhibition of miR-155 by cobomarsen, an oligonucleotide inhibitor of miR-155, slowed DLBCL tumor cell growth [360]. Selective inhibition of miR-155 function specifically inhibits the growth of lymphoblastoid cell lines and the DLBCL cell line IBL-1 [240]. Chronic active BCR signaling plays a pathogenetic role in ABC DLBCL [14]. The activation of ‘chronic’ or ‘tonic’ BCR signaling in lymphoma B cells can be influenced by a specific immunoglobulin structure, the expression and mutations of adaptor molecules (like GAB1, BLNK, GRB2, and CARD11), the activity of kinases (like LYN, SYK, and PI3K) or phosphatases (like SHIP-1, SHP-1, and PTEN) and levels of oncogenic miRs, predominantly miR-155-5p [379].

#### 4.1.5. MicroRNA-148a Maintains Survival of Immature B Cells

miR-148a has been identified as a critical regulator of B cell tolerance and autoimmunity [380]. Elevated miR-148a-3p expression impaired B cell tolerance by promoting the survival of immature B cells after engagement of the BCR by suppressing the expression of growth arrest- and DNA-damage-inducible gene α (*GADD45A*), the PI3K inhibitor PTEN, the pro-apoptotic protein BIM as well as the TNF receptor subfamily, member 1B (*TNFRSF1B*) [380]. Of note, *TNFRSF1B* is also a target of miR-125b-5p and let-7-5p. GADD45α regulates G2/M checkpoints and the entry to mitosis by dissociating the CDK1/cyclin B kinase complex [381]. Moreover, GADD45α participates in DNA repair by binding and activating the proliferating cell nuclear antigen, a key element in excision base repair [382]. Interestingly, apoptosis induced by anti-MEK small molecule AZD6244 was BIM-dependent in DLBCL cells [383].

Recent evidence indicates that miR-148a-3p promotes plasma cell (PC) differentiation via targeting GC transcription factors BTB and CNC homology 2 (*BACH2*) and microphthalmia-associated transcription factor (*MITF*) [384]. For differentiation of mature activated B cells into long-lived antibody-secreting PCs, BACH2 and MITF, are essential, as they delay premature differentiation of GC B-cells by repressing BLIMP1 and IRF4 [384]. Therefore, BACH2 and MITF expression must be attenuated in activated B cells to allow terminal PC differentiation. Notably, high BACH2 expression in GCB and non-GCB DLBCL exhibited shorter 3-year overall survival compared to DLBCLs with low BACH2 expression [385]. The analysis of miR-148a-deficient mice revealed reduced serum Ig, decreased numbers of newly formed plasmablasts, and reduced CD19-negative, CD93-positive long-lived plasma cells [386].

Thus, miR-148a may exert dual functions in B cell development, i.e., promoting the survival of immature B cells but also driving terminal differentiation of PCs. According to a recent systematic review by Yazdanparast et al. [273], exosomal miRs are not only involved in critical pathobiochemical mechanisms in DLBCL but are also useful for diagnosis and prognosis in DLBCL and are promising therapeutic tools and predictors of response to therapy.

### 4.2. Potential Uptake of Bovine Milk-Derived Exosomes by B Cells

Pasteurized milk is consumed in high quantities in the US and northern European countries. Swedes, for instance, who per capita consumed 98 L milk in 2018, primarily drink pasteurized milk [387]. In contrast to ultraheat treatment (UHT) (135–150 °C, 1–10 s), pasteurization (72 °C for 15 s) is a gentle thermal processing [388]. Pasteurization in contrast to UHT preserves milk exosomes and substantial amounts of their miRs [389,390]. Compared to raw cow’s milk, pasteurization and homogenization reduces the concentration of milk EVs [389]. However, it should be taken into account that cow’s milk contains 10^12^–10^14^ exosomes per mL [391] and thus delivers 10- to 1000-fold higher exosome numbers/mL compared to human milk. Bovine milk exosomes survive the harsh conditions in the gastrointestinal tract [392,393].

Bovine milk-derived exosomes (MDEs) are taken up by endocytosis as shown in intestinal and vascular epithelial cells [394,395] and breast cancer cells [396]. There is compelling evidence that bovine MDEs and their miRs are bioavailable [397,398,399,400], reach the systemic circulation [387], and enter the cells and peripheral tissues [401,402,403,404,405,406,407,408]. There is ample evidence that human and bovine MDEs are taken up by human peripheral blood mononuclear cells (PBMCs) [409], including macrophages [410,411,412] and T cells [413]. Wolf et al. [394] showed that the uptake of bovine MDEs is mediated by endocytosis and depends on cell and exosome surface glycoproteins in human and rat intestinal cells. In accordance, Kusuma et al. [395] observed that bovine MDEs are taken by endocytosis in vascular endothelial cells. Curcumin encapsulated in MDEs resists human digestion and possesses enhanced intestinal permeability in vitro [414]. Recently, González-Sarrías et al. [396] reported the uptake of curcumin and resveratrol-loaded bovine MDEs in MCF-7 breast cancer cells primarily was dependent on clathrin-mediated endocytosis. Caner et al. [286] recently showed that normal B cells are able to internalize exosomes derived from DLBCL resulting in changes in miR expression profiles specific to lymphoma subtypes. B cells exhibit a very active endocytic machinery. The best-characterized mechanism of BCR internalization is clathrin-mediated endocytosis [415,416,417,418,419,420]. Recent evidence indicates that class A scavenger receptor-1/2 facilitates the uptake of bovine MDEs in murine bone marrow-derived macrophages and C57BL/6J mice [421]. Baos et al. [422] described the presence of class A scavenger receptor 1 (CD204) in several cell populations of PBMCs, specifically in T and B lymphocytes, as well as monocytes. Notably, an increase in CD204^+^ cell numbers was associated with poor clinical outcome in DLBCL of the central nervous system [423]. Betker et al. [402] suggest that the absorption of bovine MDEs from the gastrointestinal tract is mediated via neonatal Fc receptor (FcRn). Of note, the expression of FcRn has been demonstrated on human B cells and murine primary B cells [424,425,426]. Thus, several potential mechanisms may mediate the uptake of bovine MDEs by B cells and B cell precursors.

### 4.3. Bovine Milk Exosome-Derived MicroRNAs

#### 4.3.1. MicroRNAs of the Let-7 Family

The most abundant miRs of MDEs are highly conserved between mammals [427,428]. Interestingly, several abundant miRs including the let-7 family members let-7a, let-7b, let-7f, and miR-148a are shared between species. Moreover, milk-derived miRs have been implicated in immune-related functions and regulation of cell growth and signal transduction [428,429]. Interestingly, the bta-let-7 family members let-7a, let-7b, let-c, let-7d, let-7f, let-7g, and let-7i are expressed in bovine MDEs [411,428,429,430,431,432,433,434]. Notably, in vivo treatment with bovine MDEs resulted in an increase in let-7a and miR-148a and expression in hepatic stellate cells [435]. Thus, the potential uptake of bovine MDEs by DLBCL cells may enhance cellular let-7 abundance promoting epigenetic down-regulation of BLIMP1 [325].

#### 4.3.2. MicroRNA-125a and MicroRNA-125b

A large quantity of bovine milk bta-miR-125b (∼10^9^–10^10^ copies/300 mL milk) withstood digestion under simulated gastrointestinal tract conditions [383]. Of note, bovine milk bta-miR-125b that resists digestion was associated with exosomes [411]. However, the majority of digested dairy milk bta-miR-125b was associated with EVs sedimenting at a centrifugation speed lower than that for exosomes [392]. Comparative expression profiles of immune-related miRs of buffalo milk nanovesicles (50–200 nm) confirmed the expression of miR-125b, miR-155, and miR-21, respectively [436]. Notably, both human and bovine miR-125b-5p exhibit an identical sequence (mirbase.org). Importantly, miR-125b targets *TP53* [329,330] and *TNFAIP3* (A20) [288]. *TNFAIP3* (A20) is a tumor suppressor gene in lymphomas [188,437] and is frequently inactivated in B-cell lymphomas [438]. A20 inhibits NF-κB signaling [439,440,441]. Evidence has been presented that epigenetic termination of TNFAIP3 function by miR-125a and miR-125b could strengthen and prolong NF-κB activity, which promotes DLBCL lymphomagenesis [442]. Shome et al. [443] identified bta-miR-125a as one of the most abundant miRs of commercial milk related to immunological roles that were not affected by pasteurization, homogenization, or heat treatment. The seed sequence of hsa-miR-125a-5p and bta-miR-125a-5p are identical. The fine-tuning of TNFAIP3 levels mediated by miR-125 expression had a striking impact on K-63 ubiquitination of TRAF2 and RIP1, IκBα degradation, p65 nuclear accumulation, and transcription of NF-κB target genes [443]. These events defined an antiapoptotic profile compatible with constitutive NF-κB activation and enhanced the fitness of B lymphoma cells.

Together, miRs of the miR-125 family play a key role in immune cell activation and oncogenesis, and constitutive activation of the NF-κB pathway in DLBCL [442,443,444]. In accordance, exosome-derived miR-125b-5p reduced the sensitivity of DLBCL to chemotherapy and rituximab [277,278]. Furthermore, the miR-125 family has been implicated as an aggravating factor in several other B cell malignancies [444].

#### 4.3.3. MicroRNA-21

Bovine exosomal miR-21 was not affected by digestion in vitro [396] and was stable under different household storage conditions, indicating that miR-21 could be biologically available to milk consumers [436]. Notably, human and bovine miR-21-5p share identical nucleotide sequences [445]. Mutai et al. [446] confirmed miR-21-5p as a component of exosomes derived from pasteurized commercial cow’s milk and showed that miR-21-5p plasma levels significantly increased by 147% 3.2 h after consumption of 1 L commercial milk by healthy human volunteers [446]. miR-21 is an abundant signature miR of cow’s milk [427,430,447] and is a component of bovine MDEs [268,446,448]. In addition to miR-125b-5p, miR-21-5p targets and inhibits *TNFAIP3* [449,450]. Furthermore, miR-21 plays an oncogenic role by targeting *FOXO1* [451,452,453] and *PTEN* [454,455,456], thereby activating the PI3K/AKT/mTORC1 pathway in DLBCL [350]. B-cell lymphoma/leukemia 11B (*BCL11B*) is another direct target of miR-21 [457]. Its loss of function in mice contributes to lymphomagenesis [458,459]. Activation of eIF4F by increased mTORC1 signaling has a direct role in lymphomagenesis due to increased synthesis of oncogenes that are dependent on enhanced eIF4A RNA helicase activity for translation [460]. AKT activation leads to the degradation of programmed cell death 4 (PDCD4), which inhibits the translation initiation factor EIF4A, an RNA helicase that catalyzes the unwinding of secondary structure at the 5′ UTR of mRNAs [461] and can alter eIF4F complex formation [460]. Notably, *PDCD4* is also epigenetically downregulated by miR-21 [239,354,355]. It is well-accepted that miR-21 is frequently upregulated in DLBCL [342,343,344]. Moreover, insulin-like growth factor (IGF)-binding protein-3 (*IGFBP3*) is a miR-21 target gene promoting glioblastoma tumorigenesis [462]. Thus, miR-21 enhances levels of free IGF-1. The majority of reported targets of miR-21 are tumor suppressor genes and inhibitors of apoptosis [239,463,464,465] (Figure 2).

#### 4.3.4. Exosomal MicroRNA-21 Exposure and M2 Macrophage Polarization

The tumor microenvironment (TME) of DLBCL and its role in DLBCL pathogenesis and patient survival is a matter of recent research [466]. The TME consists of T and B lymphocytes, tumor-associated macrophages (TAMs), myeloid-derived suppressor cells (MDSCs), cancer-associated fibroblasts (CAFs), and other components [467]. The crosstalk between malignant B cells and immune cells in the lymphoma TME is highly complicated and might be affected by often interconnected intrinsic and/or extrinsic mechanisms that ultimately can lead to immune escape [467]. M2-polarized macrophages are implicated to contribute to DLBCL progression and poor patient outcome [466,468]. Tumor-associated macrophages (TAMs) comprise an important part of the TME, are predominantly M2-polarized, and play a key role in DLBCL progression [469,470,471,472]. Compelling evidence in various cancer entities underlines the impact of miR-21-5p in driving M2 polarization of macrophages and TAMs [473,474,475,476], whereas inhibition or depletion of miR-21 reversed M2 polarization back to the M1 phenotype [477,478,479]. Remarkably, miR-21 depletion in macrophages promotes tumoricidal polarization and enhances PD-1 immunotherapy [477].

Exosome-mediated crosstalk between tumors and TAMs involves the traffic of miR-21, which is the recent focus on molecular oncology [480]. DLBCL-derived EVs and exosomes are internalized by macrophages and can induce M2 macrophage polarization and thus contribute to tumor progression [481,482]. Importantly, tumor-cell-derived EVs and exosomes transfer miR-21-5p to monocyte/macrophages inducing the M2-phenotype [483,484,485,486,487]. In fact, it has been demonstrated that miR-21-5p-containing exosomes were engulfed by CD14^+^ human monocytes, suppressing the expression of M1 markers and increasing that of M2 markers [484]. The transcription factor STAT3 is a known enhancer of miR-21 expression [488,489,490]. Preclinical evidence indicates that the STAT3 inhibitor pacritinib could overcome temozolomide resistance via downregulating miR-21-enriched exosomes from M2 glioblastoma-associated macrophages [491]. It has been demonstrated in colon cancer that TAM secrete exosomes containing miR-21-5p and miR-155-5p promoting cell migration and invasion [492]. In hepatocellular carcinoma (HCC), miR-21-5p expression was upregulated in M2 macrophage-derived EVs, which carried miR-21-5p into HCC tissues [493]. Of note, M2 macrophage-derived EVs promoted the depletion of CD8^+^ T cells in HCC via the miR-21-5p/YOD1/YAP/β-catenin axis [493]. In pancreatic cancer, M2 macrophage-derived exosomal miR-21-5p stimulated differentiation and activity of pancreatic cancer stem cells via targeting Kruppel-like factor 3 (KLF3) [494].

Taken together, compelling evidence supports the view that tumor cell-derived exosomal miR-21-5p promotes macrophage M2 polarization within the TME. M2-polarized TAMs themselves secrete miR-21-5p enriched exosomes into the TEM further stimulating tumor cell growth. In this scenario, it is of critical concern that bovine MDEs, which are enriched in miR-21-5p [227,446,448] are taken up by human PBMCs [403] and human macrophages [411]. Thus, dietary MDE miR-21 exposure may enhance the total burden of miR-21 signaling in the TME promoting lymphomagenesis and tumor progression [334] (Figure 3).

#### 4.3.5. MicroRNA-29b

In comparison to colostrum, mature cow’s milk contains more abundant amounts of miR-29b [411]. Pasteurized and homogenized 2% fat commercial cow’s milk stored at 4 ℃ for 15 days still exhibited more than 50% of the initial miR-29b concentration detected in fresh raw cow’s milk [495]. Yu et al. [448] reported that bovine MDEs contain miR-21 and miR-29b, whereas Izumi et al. [411] detected miR-29c in bovine MDEs. Baier et al. [409] demonstrated that postprandial plasma concentrations of miR-29b increased in a dose-dependent manner after intake of 0.25, 0.5, and 1.0 L of commercial cow’s milk in healthy human volunteers. Notably, the expression of runt-related transcription factor 2 (RUNX2), which is positively regulated by miR-29b [496], increased by 31% in PBMCs after milk consumption compared with baseline [409]. Of note, the nucleotide sequence of bovine miR-29b is identical to that of human miR-29b [497]. RUNX2 is highly expressed in adherent B-NHL cells compared to cells in suspension, and knockdown of RUNX2 expression could reverse cell adhesion-mediated drug resistance. Furthermore, RUNX2 could promote the proliferation of B-NHL cells [498]. Hines et al. [499] reported that miR-29s in murine B lymphocytes regulate the BCR-PI3K signaling cascade by dampening PTEN expression and that loss of this miR cluster results in increased apoptosis as well as defects in B cell terminal differentiation. Remarkably, the miR expression profile of U2932 DLBCL cells transfected with EBV-encoded latent membrane protein 1 (LMP1) exhibited significant upregulation of miR-29a, miR-29b, and miR-29c [500]. However, in this experimental model system of DLBCL, miR-29b downregulated the expression of the T-cell leukemia gene 1 (*TCL1*) [501], which exhibits oncogenic activity in distinct classes of B cell lymphoma [501,502].

#### 4.3.6. MicroRNA-155

The quantity of immune-related miR-155 of colostrum is 5.4-fold higher than in raw mature cow’s milk [429]. Levels of miR-155 of HTST (high temperature for short time, 75 °C for 15 s) milk was 3.8-fold higher for miR-155 compared to LTLT (low temperature for a long time, 63 °C for 30 min) cow’s milk [429]. The lowest miR-155 levels have been detected in UHT (ultra-heat treatment 120–130 °C for 0.5–4 s) [429]. It appears that the common procedure of cow’s milk pasteurization (75 °C for 15 s) does not lead to a significant destruction of MDE miR-155. Mutai et al. [446] detected miR-155-5p in bovine MDEs. Notably, the stability of miR-155 in raw cow’s milk was not affected by incubation at 37 °C for 1 h without treatment, incubation at 37 °C for 1 h with RNase (10 U/mL RNase A and 400 U/mL RNase H), exposure to a low pH (pH 2), or treatment with detergent (1% Triton X-100) compared to untreated bovine milk [503]. Remarkably, pretreatment of IEC-6 cells with bovine MDEs increased intracellular levels of miR-155 [504]. It has recently been shown that the primary oncogene of EBV, latent membrane protein 1 (LMP1), upregulates the expression of miR-155 in EBV^+^ B cell lymphoma cell lines and associated exosomes targeting the anti-apoptotic protein FOXO3A [505]. Notably, resistance to Bruton’s tyrosine kinase (BTK) inhibition by ibrutinib was associated with the downregulation of FOXO3A and PTEN levels and activation of AKT [506].

In fact, high levels of EV-associated miR-155 were found to correlate with chemotherapy resistance in several common cancers including DLBCL [507]. In patients with ABC DLBCL, Zare et al. [378] showed that exosomal miR-155 expression was upregulated in refractory/relapsed patients compared to responsive patients and patients receiving R-CHOP. Therefore overexpression of exosomal miR-155 in refractory/relapsed patients might be associated with more aggressive disease, poor response to R-CHOP therapy, and adverse prognosis [377]. In HCV-infected patients with rheumatoid arthritis, rituximab declined exosomal miR-155 concentrations [508]. Notably, a transfer of exosomal KSHV-miR-K12-11 (miR-K12-11) that has an identical seed sequence as miR-155, from KSHV-infected BCBL-1 and BC-1 lymphoma lines to T cells has been reported [509].

Recent evidence indicates that miR-155 enhanced lymphoma cell programmed death ligand 1 (PD-L1) expression, whereas PD-L1 blockade particularly retarded miR-155-overexpressing tumor growth consistent with the maintenance of CD8^+^ T cells and their function [510]. Targeting the polarization of tumor-associated macrophages and modulating miR-155 expression might be a new approach to treating DLBCL in the elderly [511].

#### 4.3.7. MicroRNA-148a

miR-148a is a signature miR of commercial cow’s milk [447] and human milk and milk EVs [512,513]. In contrast to the boiling or UHT of cow’s milk, pasteurization and homogenization did not result in losses of miR-148a-3p compared to reeds of raw cow’s milk [514]. Benmoussa et al. [515] found that the bulk of bovine milk miRs including bta-miR-148a and bta-miR-125b sediment at 12,000 g and 35,000 g. Nevertheless, miR-148a is an abundant non-coding RNA detected in human [516,517], bovine [411], and porcine MDEs [518]. Human and bovine miR-148a-3p shares identical nucleotide sequences [519]. Recent evidence underlines that oral administration of exosomal bovine miR-148a-3p is bioavailable and reaches distant target tissues of C57BL6/mice [400]. Bovine MDE miR-148a-3p targets *DNMT1* [427], the key maintenance DNA methyltransferase linking milk signaling to epigenetic regulation [519]. Guo et al. [517] recently demonstrated that human MDE miR-148a-3p directly targets *TP53*. Thus, MDEs via transfer of miR-148a-3p and miR-125b-5p to recipient cells may down-regulate the expression of p53 [520]. Of importance, p53 regulates the baseline expression of key genes involved in cell homeostasis such as *FOXO1*, *PTEN*, *SESN1*, *SESN2*, *AR*, *IGF1R*, *BAK1*, *BIRC5*, and *TNFSF10* [521]. In fact, aberrations of the *TP53* gene and dysregulation of the p53 pathway are regarded to be important in the pathogenesis of DLBCL [327]. Remarkably, *TP53* loss led to an upregulation of programmed death ligand 1 (PD-L1) cell surface expression and secretion of EVs with EV-bound PD-L1 by lymphoma cells [522]. *PTEN* is a direct target of miR-148a-3p [523,524,525]. In fact, Reif et al. [526] demonstrated that human MDEs via miR-148a-3p down-regulated PTEN expression after incubation of normal fetal colon epithelial cells with MDEs. It is noteworthy that PTEN deficiency or PTEN loss are common adverse features in the pathogenesis of DLBCL [527,528,529,530].

BCL6 is a transcriptional repressor critically involved in the development and maintenance of GCs and lymphomagenesis [531]. Mutations and translocations leading to the sustained activity of BCL6 promote the development of GC-derived lymphomas [531]. Epigenetic mechanisms also lead to B cell hyperactivation [532,533]. In fact, BCL6 activity is modified by epigenetic mechanisms. As mentioned earlier, AICDA and DNMT1 form a complex to maintain methylation of the *BCL6* promoter thereby inhibiting its expression. Loss of either AICDA or DNMT1 causes instability of the AICDA-DNMT1 complex resulting in its disassociation from the *BCL6* promoter enhancing BCL6 expression in DLBCL [28] (Figure 2).

Importantly, *DNMT1* is a direct target of MDE miR-148a-3p [235,526,534], whereas *AICDA* is targeted by miR-155-5p [535,536,537] and miR-29b [538]. Thus, continued consumption of pasteurized milk with a continuous transfer of MDE miR-148a, miR-155, and miR-29b via suppressing the AICDA-DNMT1 complex on the *BCL6* promoter may maintain high expression of the oncogene BCL6 promoting lymphomagenesis.

Alles et al. [539] recently reported that miR-148a-3p impairs RAS/ERK signaling in B lymphocytes by targeting the Son of Sevenless 1 (SOS1) and SOS2 proteins. Notably, increased expression of miR-148a-3p reduced the expression of ERK1/ERK2 [539]. ERKs are essential for the differentiation of B cells into antibody-producing plasma cells and induce the expression of BLIMP1, a transcriptional repressor and “master regulator” of plasma cell differentiation [294,540,541,542]. In contrast, it has also been shown that miR-148a promotes plasma cell differentiation via the targeting of the BLIMP1 and IRF4 repressors MITF and BACH2 [384]. Premature expression of miR-148a by retroviral transduction favored plasmablast differentiation and the survival of in vitro activated primary murine B cells [384].

There is recent interest in PD-1/PD-L1 pathway blockade in patients with DLBCL [384,543,544,545,546,547,548,549]. Remarkably, Ashizawa et al. [550] demonstrated that PD-L1 is a direct target of miR-148a-3p in colorectal cancer cells. However, with the exception of the miR-148a/152 family, there is no other MDE-associated miR known to target PD-L1 [551]. Thus, miR-148a-3p may exert both tumor-promoting and protective effects in the pathogenesis of DLBCL (Table 3).

## 5. Discussion

Components of commercial cow’s milk activate mTORC1 signal transduction [53,54] and MDE miR-based epigenetic regulatory signaling [526], which share common pathways upregulated in DLBLC (Figure 1 and Figure 2). The transcriptional repressor BCL6 controls a large transcriptional network that is required for the formation and maintenance of GCs. GC B-cells represent the normal counterpart of most human B-cell lymphomas, which are often characterized by upregulated BCL6 expression and BCL6-mediated pathways [552]. BCL6 originally identified as encoded by a frequently translocated locus in DLBCLs [553], serves as a master regulator of the GC reaction [296,297]. BCL6 plays a central role in normal B cell development as well as lymphomagenesis [554]. Importantly, recent evidence indicates that BCL6 expression is regulated by epigenetic mechanisms involving the complex formation of AICDA and DNMT1 at the *BCL6* promoter, which attenuates BCL6 expression through DNA methylation [28]. Human breastmilk and cow’s milk mediated transfer of MDE miRs that attenuate the expression of DNMT1 (miR-148a) and AICDA (miR-155, miR-29b) may thus enhance BCL6 expression promoting GC formation. This may represent a physiological effect on the newborn infant during the period of lactation. However, there is only limited information on the impact of breastmilk and its components on the epigenetic programming of immune function and immune development in early life [555]. Notably, BCL6 is critical for the development of a diverse primary B cell repertoire [556]. Importantly, there is intensive reciprocal and antagonistic crosstalk between BCL6 and BLIMP1 [557]. Lymphocytes with higher expression of BCL6 exhibit greater proliferative capacity but less secretory capacity, whereas lymphocytes with higher expression of BLIMP1 exhibit lower proliferative capacity and greater secretory capacity [556]. Of note, BLIMP1 is also regulated by miRs [321]. Especially, let-7 and miR-125b, which are overexpressed in DLBCL, suppress BLIMP1 expression [324,325,326]. BLIMP1 drives terminal differentiation in B cells and promotes plasmablast formation and antibody secretion [295]. The abundant milk-derived miRs of the let-7 family as well as miR-125b via targeting *PRDM1*, the gene encoding BLIMP1, may help to restrict unnecessary plasmablast and plasma cell formation and antibody synthesis during the postnatal period of lactation. During lactation, human milk provides abundant specific antibodies that protect the infant [558]. This is necessary because neonates exhibit an immature immune system and their immune activities are different from the activities of the adult immune system [559,560]. Remarkably, miRs are involved in the regulation of the immature neonatal immune system [560]. Neonatal and adult B cells are comparable in their proliferative responses to cooperative cell interactions. However, a marked deficiency in the ability of neonatal B cells to mature to immunoglobulin secretion was observed [561]. Apparently, “early” B lymphocytes are intrinsically defective in their ability to secrete immunoglobulin upon cooperative induction, whereas they show full competence to expand clonally [561]. Milk-derived miR signaling may epigenetically enhance the expression BCL6 expression and reduce the expression of BLIMP1 during the period of lactation thus promoting B lymphocyte proliferation and suppressing B cell differentiation and function. Weaning, the termination of breastmilk, maternal antibodies, and MDE miRs, may initiate an “epigenetic switch” attenuating BCL6 and enhancing BLIMP1 expression required for B cell differentiation to plasma cells and own antibody secretion by the infant [557]. This scenario is in accordance with the MDE miR-mediated switch of proliferating pancreatic β-cells to differentiated insulin-secreting matured β-cells after weaning [562,563]. Continued consumption of commercial cow’s milk via transfer of highly conserved MDE miRs may not only de-differentiate the pancreatic β-cell back to the neonatal phenotype but may also over-stimulate BCL6-driven B cell proliferation, a potential driving force in the pathogenesis of DLBCL (Figure 4).

Key new findings implicate DNA methylation heterogeneity as a core feature of DLBCL underlining the role of epigenetic dysfunction on lymphomagenesis [20,21,22,26,564,565,566,567]. Current findings highlight the potential role of miRs (lymphomiRs) as important factors in lymphomagenesis [568]. At present, it is not possible to oversee the potential contribution of bovine MDE miRs and their targets in epigenomic dysregulations of DLBCL and its subtypes (ABC, GCB) during its various stages of lymphomagenesis. Nevertheless, the presented epidemiological and mechanistic insights link the consumption of cow’s milk and bovine MDE miRs to the pathogenesis of DLBCL.

It is noteworthy that the distribution and consumption of pasteurized cow’s milk is a new human behavior in relation to the human history of milk consumption since ancient times [387]. MDEs and MDE miRs survive pasteurization, whereas MDE and milk miRs are destroyed by boiling or UHT [389,390] and are attacked by fermentation [448]. Thus, future epidemiological studies relating to milk consumption and DLBCL risk have to consider the methods of thermal milk processing. Whereas MDE miRs may function as a pathogenic factor in DLBCL, MDE miRs may exert beneficial effects in the prevention of colorectal cancer (CRC). The consumption of cow’s milk has been related to a decreased risk of CRC [569,570,571,572]. In fact, bovine MDEs exhibited anti-inflammatory effects in murine models of colitis [573,574]. Notably, miR-148a-3p and MDE miR-27b exert tumor suppressive functions in CRC [575,576,577].

## 6. Conclusions

Whereas MDEs and their miRs are of key importance for neonatal development [578,579], MDEs may exert pathogenic effects in adults [579]. In this light, bovine MDEs should be regarded as pathogens that have to be excluded from the human food chain [580,581]. With regard to DLBCL pathogenesis, we are concerned about recent recommendations promoting bovine miR-155-rich colostrum for “immune-nutrition” in elderly subjects [582] as well as bovine MDEs as “health-improving bioactive ingredients” in the context of human nutrition [583]. Persistent exposure to bovine miR-155-rich colostral MDEs may thus function like EBV-driven exosomal overexpression of miR-155 inducing lymphomagenesis [505]. In addition, medical advice to include whole cow’s milk in the diet for patients with DLBCL is questionable [584]. Obviously, there is an urgent need to study milk as a biological system [585], which is the basis to understand milk’s beneficial and adverse effects on human health during the differing stages of immune system development in infancy and homeostatic maintenance of immune functions in adulthood [586]. DLBCL accounts for approximately 30% of adult lymphomas and 10% of lymphomas diagnosed before the age of 18 years [587]. Importantly, the incidence of DLBCL increases with age [588]. The average age at the initial diagnosis of DLBCL is approximately 70 years [589,590]. Unfortunately, current epidemiological studies do not report lifetime periods of cow’s milk exposure in relation to the occurrence of DLBCL. Thus, there is no information on potential vulnerable windows for cow’s milk consumption during infancy or adolescence and the later onset of DLBCL as demonstrated for prostate cancer [591]. The dietary exposure of humans to MDEs and miRs beyond the weaning period is an unnatural intervention that may represent a pathogenic risk factor for all age groups [52]. Cow’s milk-mediated transmission of oncogenic viruses is a recent matter of concern that requires further investigation. In contrast to cow’s milk and dairy products, a higher intake of green leafy vegetables and cruciferous vegetables has been associated with a lower overall risk of NHL, particularly DLBCL and follicular lymphoma [48].

## 7. Limitations

We would like to note that the majority of reported associations between milk signaling and pathogenic signaling in DLBCL are based on literature research data and mechanistic models not on direct experimental evidence. Data that clarify the potential impact of MDE miRs on B cells as well as B cell lymphoma cells are still missing and need to be explored in experimental settings and studies including human milk consumers. The recently presented epidemiological evidence linking cow’s milk consumption to the risk of DLBCL also requires more detailed data on thermal milk processing (destruction of MDE miRs by UHT or bioavailability of MDE miRs in pasteurized milk). Because only a smaller fraction of human milk consumers will develop DLBCL during their lifetime, predisposing individual factors on the patient’s side may be a greater importance. Nevertheless, milk signaling evidently aggravates signal transduction pathways and epigenetic deviations contributing to DLBCL initiation and progression. Data on epigenetic regulatory effects of breastmilk signaling in healthy newborn infants and potential epigenetics shifts after weaning are missing and appear to be of utmost importance to understanding the physiological impact of milk signaling on postnatal B cell proliferation and development. Future studies that determine the epigenetic heterogeneity in human milk consumers versus individuals who were not exposed to cow’s milk (lactose intolerance) may provide deeper insights into the milk-DLBCL relationship.

## 8. Materials and Methods

We conducted our bibliographic research exclusively via PubMed [https://pubmed.ncbi.nlm.nih.gov] between 1995–2022 using various keywords such as “B cell”, “B lymphocyte”, ”B cell proliferation“, B cell differentiation”, “malignant B cell”, “malignant B lymphocyte”, “diffuse large B cell lymphoma”, “DLBCL”, “non-Hodgkin lymphoma”, “lymphomagenesis”, lymphocyte-induced maturation protein 1”, “BLIMP1”, “PR domain-containing protein 1”, “PRDM1”, “B cell lymphoma 6”, “BCL6”, “milk”, “cow’s milk”, “dairy”, “milk exosome”, “extracellular vesicle”, “EV”, “microRNA”, “miRNA”, “miR”, “exosomal miR”, “epigenetic”, and “epigenetic regulation”. We also analyzed papers based on the bibliographic references cited by the studies found on PubMed during our search. Whenever possible, we selected the most recent and comprehensive reviews on the topic in question. All selected articles were written in English.

## Figures and Tables

**Figure 1 ijms-24-06102-f001:**
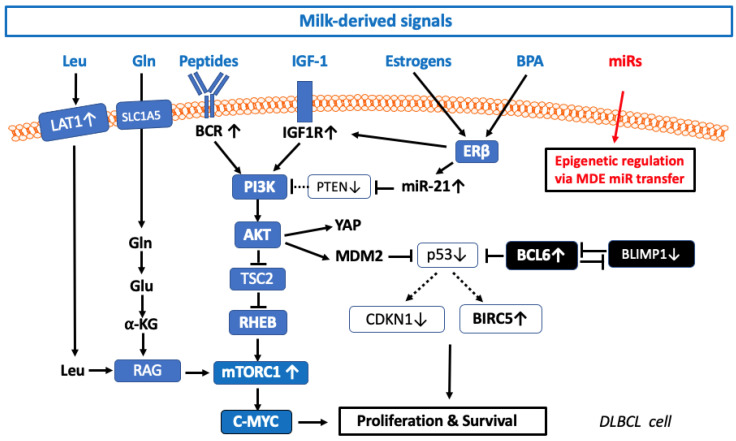
Potential synergism of milk-induced phosphatidylinositol-3-kinase (PI3K)-AKT-mechanistic target of rapamycin complex 1 (mTORC1) signaling and overstimulated PI3K-AKT-mTORC1 signaling in diffuse large B-cell lymphoma (DLBCL). B-cell receptor (BCR) is commonly activated in DLBCL and stimulates the mTORC1 pathway. Milk-derived peptides are known agents interacting with BCR. Insulin-like growth factor 1 (IGF-1)/IGF-1 receptor (IGF1R) signaling is also activated in DLBCL. Milk contains IGF-1 and stimulates IGF-1 synthesis in human milk consumers. IGF-1/IGF1R signaling activates mTORC1. Expression or function of phosphatase and tensin homolog (PTEN) is commonly suppressed in DLBCL and is also targeted by microRNAs (miRs), especially miR-21. DLBCL cells frequently overexpress L-type amino acid transporter 1 (LAT1), which plays a crucial role in intracellular uptake of leucine (Leu), which via Ragulator (RAG) activates mTORC1. Leu is an abundant amino acid of milk proteins. Glutamine (Gln) is also enriched in milk proteins. The glutaminolysis pathway is activated in DLBCL cells. The end product α-ketoglutarate (αKG) via RAG as well activates mTORC1, which finally promotes the translation of the oncogene C-MYC, the key driver of cell proliferation. The activated kinase AKT stimulates YES1-associated transcriptional regulator (YES) of the HIPPO pathway found to be activated in DLBCL. Estrogen receptor β (ERβ) was also found to be overexpressed in subtypes DLBCLs. ERβ induces the expression of IGF1R and miR-21-5p. Milk-derived estrogens and milk contamination with bisphenol A (BPA) may contribute to ERβ signaling in DLBCL. AKT-mediated activation of mouse double minute 2 (MDM2) promotes the proteasomal degradation of p53, the guardian of the genome. p53 regulates the transcription of cyclin-dependent kinase inhibitor 1A (*CDKN1A*; p21) and of the anti-apoptotic protein survivin (*BIRC5*). TP53 is frequently downregulated in DLBCL. The oncogene B cell lymphoma 6 (BCL6) is commonly upregulated in DLBCL and functions as a negative regulator of both p53 and B lymphocyte-induced maturation protein 1 (BLIMP1, *PRDM1*). As such, milk-derived exosomal microRNAs play a key role in epigenetic regulation.

**Figure 2 ijms-24-06102-f002:**
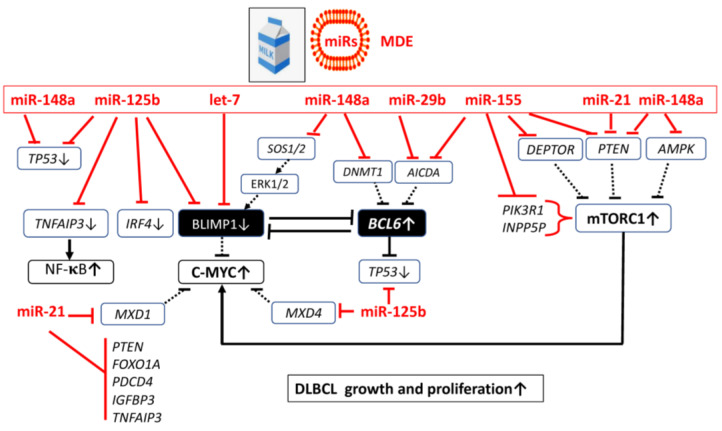
Illustrated mechanistic role of bovine milk-derived exosomes (MDE) and their microRNAs (miRs) in epigenetic signaling in diffuse large B-cell lymphoma (DLBCL). The key oncogene upregulated in most DLBCL is B cell lymphoma 6 (BCL6) activating the germinal center reaction and promoting B cell proliferation. The expression BCL6 is suppressed by a complex of DNA methyltransferase 1 (DNMT1) and activation-induced cytidine deaminase (encoded on *AICDA*), which catalyzes BCL6 promoter methylation. DNMT1 is a target of miR-148a, the most abundant miR of human and bovine milk and their MDEs. AICDA is targeted via miR-155 and miR-29b, abundant immune-regulatory miRs of human and bovine milk. MDE miRs may thus enhance BCL6 expression promoting B cell proliferation. BCL6 is a critical inhibitor of B lymphocyte-induced maturation protein 1 (BLIMP1, encoded on *PRDM1*), the key driver of B cell differentiation. miR-148a via suppression of SOS RAS/RAC guanine nucleotide exchange factor 1 and 2 (SOS1/2) inhibits ERK1/2-mediated expression of BLIMP1. Thus, miR-148a indirectly increases the ratio of BCL6/BLIMP1. BLIMP1 is directly suppressed via the let-7 family of miRs and miR-125b, further abundant miRs of human and bovine milk and their MDEs. Reduced expression of BLIMP1 enhances C-MYC expression, a key oncogene driving cell proliferation. C-MYC translation is dependent on the mechanistic target of rapamycin complex 1 (mTORC1). Negative regulators of mTORC1 are DEP domain-containing protein 6 (DEPTOR encoded on *DEPDC6*), phosphatase and tensin homolog (PTEN), and AMP-activated protein kinase (AMPK), which are targeted via abundant miR-155, miR-21 an miR-148a, respectively. In addition, miR-21 targets MAX dimerization protein 1 (*MXD1*), whereas miR-125b targets MAX dimerization protein 4 (*MXD4)*, leading to sustained C-MYC activation. In addition, miR-148a and miR-125b target *TP53,* the key transcription factor of a multitude of tumor suppressor genes. miR-125b also targets interferon regulatory factor 4 (IRF4), a well-defined transcription factor obligatory required for the terminal differentiation of B cells to plasma cells. A further target of miR-125b is tumor necrosis factor alpha-induced protein 3 (*TNFAIP3*), also known as A20. TNFAIP3 is a negative regulator of nuclear factor kappa B (NF-κB) signaling, which is upregulated in most cases of DLBCL. Notably, miR-21 also targets *TNFAIP3* as well as forkhead box O A1 (*FOXO1A*), programmed cell death 4 (*PDCD4*), and insulin-like growth factor-binding protein 3 (*IGFBP3*). Thus, the most abundant milk-derived miRs maintain mTORC1 activation, and promote the expression of BCL6 and NF-κB, but suppress cell cycle inhibitor p53 and transcription factors BLIMP1 and IRF4 involved in B cell differentiation and plasma cell and antibody formation. This signaling scenario may be useful for neonatal B cell proliferation during the period of lactation when high-affinity antibodies educated by the maternal immune system are provided to the infant via breastfeeding. However, persistent epigenetic milk miR signaling via continued exposure to bovine MDE-transferred miRs provided by the consumption of pasteurized cow’s milk may promote BCL6/BLIMP1-dependent lymphomagenesis.

**Figure 3 ijms-24-06102-f003:**
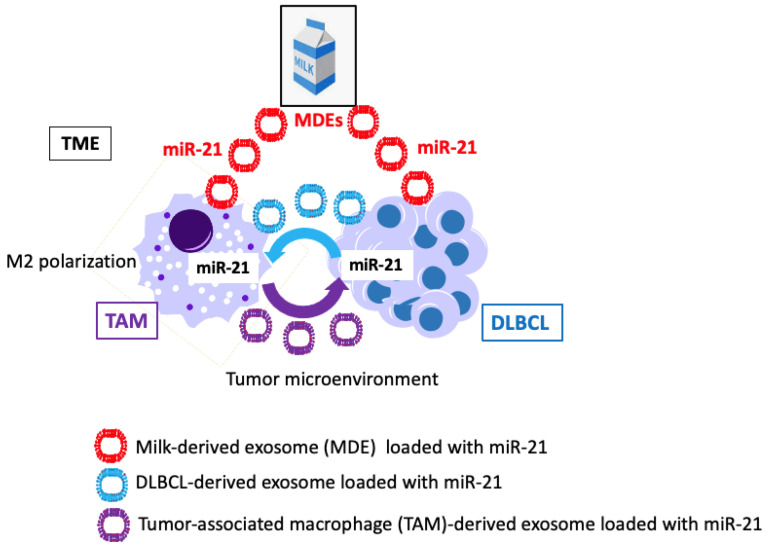
Exosome crosstalk in the tumor microenvironment (TME) of diffuse large B-cell lymphoma (DLBCL). DLBCL cells secrete microRNA-21 (miR-21)-enriched exosomes into the TME to tumor-associated macrophages (TAM), which internalize these tumor-derived exosomes and promote macrophage M2 polarization, favoring tumor growth and progression. TAMs also secrete miR-21-enriched exosomes to DLBCL cells further promoting lymphomagenesis. Bovine milk-derived exosomes (MDEs) also transport miR-21 to recipient cells that may reach TAMs and/or DLBCL cells further augmenting miR-21-mediated oncogenic signaling.

**Figure 4 ijms-24-06102-f004:**
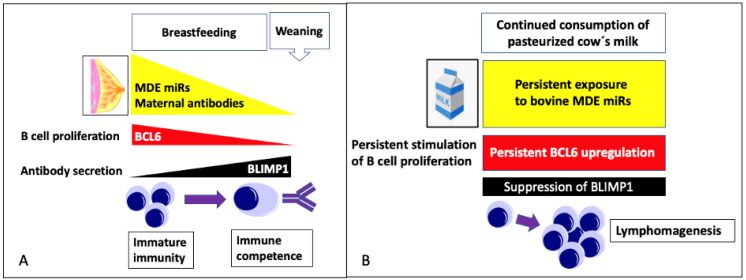
(**A**) Physiological interaction of human milk-derived exosomes (MDEs) and their microRNAs (miRs) in B lymphocyte proliferation via epigenetic upregulation of B cell lymphoma 6 (BCL6) and suppression of B lymphocyte-induced maturation protein 1 (BLIMP1), which suppresses B cell differentiation and antibody secretion. The neonatal immune system is immature and neonatal B cells are in a hyperproliferative undifferentiated state. The majority of specific antibodies are of maternal origin and are supplied via breastfeeding. Weaning, the physiological fading of MDE miRs, induces a key switch in milk-driven epigenetic regulation enhancing BLIMP1 expression and promoting antibody secretion by the infant’s matured immune system. (**B**) Persistent consumption of pasteurized cow’s milk maintains MDE miR-mediated epigenetic signaling promoting BCL6 expression and suppressing BLIMP1 expression, thus maintaining the hyperproliferative neonatal phenotype. Sustained proliferation is a hallmark of cancer, in this setting lymphomagenesis.

**Table 1 ijms-24-06102-t001:** Recent epidemiological evidence for milk and dairy product consumption related to increased risk for B-cell lymphoma and DLBCL.

Study Characteristics	Associated Risk	References
Meta-analysis 14 case-control studies and two cohort studies	Milk consumption is related to increased risk of non-Hodgkin lymphoma (NHL) RR = 1.49 (95% CI: 1.08–2.06); risk of NHL increased by 5% and 6% for each 200 g/day increment in total dairy product and milk consumption, respectively	[49]
An exposome-wide analysis based on the European Prospective Investigation into Cancer and Nutrition Study (n = 475,426)	Positive association between dairy intake and risk of B-cell lymphoma and DLBCL	[51]
Prospective China Kadoorie Biobank Study collecting ~0.5 million adults from 10 diverse (5 urban, 5 rural) areas across China during 2004–2008	Increased lymphoma risk related to dairy consumption (mainly milk) for urban HR = 1.09 (95% CI: 0.89–1.34) and rural regions HR = 1.45 (95% CI: 1.07–1.96)	[50]

**Table 2 ijms-24-06102-t002:** Comparison of biologically active compounds related to milk consumption and DLBCL-associated pathology.

Compound	Cow’s Milk Intake	References	DLBCL	References
IGF-1	Increased serum levels of IGF-1 activate mTORC1	[67,68,69,70,71,72,73,74,75,76]	Increased IGF-1/IGF1R signaling	[61,62,63,64,65]
Leucine	Rich source of leucine (10 g leucine/100 g milk protein) activates mTORC1	[92,93,94,95,96,97]	High expression of leucine transporter LAT1 associated with poor prognosis	[98,99]
Glutamine	Rich source of glutamine 9 g glutamine/100 g casein	[106,107]	Glutaminolysis activates TCA cycle and mTORC1	[101,102,103]
Milk-derived peptides	B-cell epitopes stimulate BCR signaling	[147,148,149,150,151,152,153,154,155]	Persistent BCR activation augments mTORC1 signaling	[108,109,110,111,112,113,128,129,130,131,132]
Estrogens	Increased levels in milk of persistently pregnant dairy cows	[167,168,169,170,171,172,173,174]	High ERβ expression is related to poor prognosis	[157,158,159]
Bisphenol A (BPA)	Contamination of commercial milk, weak estrogenic activity increases IGF-1/mTORC1	[201,202,203,204,205]	Lymphoma-promoting activities of BPA	[37,182,183,184,185,186]
BMMFs	Viral single-stranded DNA	[248,249]	Suspected to promote cancer induction early in life; Role in DLBCL not yet experimentally investigated	[250,256]

Abbreviations: IGF-1, insulin-like growth factor 1; IGF1R, insulin-like growth factor 1 receptor; LAT1, L-type neutral amino acid transporter 1; mTORC1, mechanistic target of rapamycin complex 1; ERβ, estrogen receptor beta; BPA, bisphenol A; BMMFs, bovine meat, and milk factors; DNA, desoxyribonucleic acid; DLBCL, diffuse large B cell lymphoma.

**Table 3 ijms-24-06102-t003:** Comparison of microRNAs (miRs) implicated in the pathogenesis of DLBCL with bioactive milk-derived microRNAs.

MiRs	DLBCL-Related miRs	References	Cow’s Milk- Derived miRs	References
let-7	Over-expression of let-7 suppresses BLIMP1 (*PRDM1*)	[324,325]	Highly conserved in milk; component of MDEs	[411,427,428,429,430,431,432,433,434]
miR-125b	Increased serum levels of exosomal miR-125b-5p in DLBCL patients associated with shorter progression-free survival time and chemo-therapy resistance; targets *TNFAIP3* (A20), the negative regulator of NF-κB; down-regulates the expression of interferon regulatory factor 4 (*IRF4*) and BLIMP1; targets *TP53*; reduces BLIMP1-mediated repression of pri-miR-21	[287,288,326,328,329,330,342]	Resistant component of milk EVs and MDEs: Postprandial increase in plasma after consumption of commercial cow’s milk in humans;	[392,411,436,446]
miR-21	Increased serum levels of patients with DLBCL miR-21; increased in exosome-enriched serum of patients with DLBCL; increased levels in DLBCL tumors; targets *MAXD1* (MAX dimerization protein 1) promoting the formation of C-MYC-MAX heterodimers that activate C-MYC expression; targets Von Hippel Lindau mRNA (*VHL*); inversely correlated with the levels of FOXO1 and PTEN in DLBCL cell lines targets programmed cell death 4 (*PDCD4*); targets *IGFBP3*; drives M2 polarization of macrophages	[239,286,329,330,331,332,333,334,335,336,337,338,339,343,347,348,349,352,460,462,471,472,473,474]	Abundant signature miR of cow’s milk; component of MDEs; targets *TNFAIP3*, *FOXO1*, *PTEN*, *BCL11B*, and *PDCD4*; promotes M2 macrophage polarization	[238,267,353,354,427,430,446,447,448,449,450,451,452,453,454,455,456,457,473,474,475,476]
miR-29s	Transfection of U2932 DLBCL cells twith EBV-encoded latent membrane protein 1 (LMP1) exhibited significant upregulation of miR-29a,b and c; miR-29b enhances *RUNX2* expression, which promotes proliferation of B-NHL cells; targets AID (*AICDA*) enhancing BCL6 expression	[496,498,538]	Component of MDEs; dose-dependent increase in plasma after intake of commercial cow’s milk; increasing RUNX2 expression in PBMCs	[409,411,448]
miR-155	Higher levels of miR-155 are present in DLBCL with an ABC phenotype; highly expressed in activated B cells and proliferating lymphoblastoid cell lines; targets the PI3K inhibitor PIK3R1; activates BCR-PI3K-AKT-mTORC1 signaling at various checkpoints via targeting *PTEN*, *DEPTOR,* and SHIP1 (*INPP5D*); EBV upregulates miR-155. Higher miR-155 plasma levels are associated with short survival; higher levels of plasma exosomal miR-155 correlate with reduced R-CHOP response; targets AID (*AICDA*) enhancing BCL6 expression, the key suppressor of BLIMP1	[28,240,241,299,358,359,360,361,362,363,364,365,366,371,372,377,378,535,536,537]	Immune-related miR of colostrum and cow’s milk; pretreatment of IEC-6 cells with MDEs increased intracellular levels of miR-155	[429,504]
miR-148a	miR-148a-3p impairs RAS/ERK signaling in B cells by targeting SOS1/2 thereby reducing ERK-mediated expression of BLIMP1; miR-148a-3p targets DNMT1; enhances the expression of BCL6, the key suppressor of BLIMP1; elevated miR-148a-3p expression impaired B cell tolerance by promoting the survival of immature B cells	[28,235,299,380,526,534,539]	Signature miR of commercial cow’s milk; component of milk EVs and MDEs; targets *DNMT1*, *TP53*, *PTEN*, *SOS1* and *SOS2*, thereby reduces ERK-mediated activation of BLIMP1; targets *MITF*, *BACH2* and *PDL1*	[384,427,447,511,515,517,526,527,539,551]

## Data Availability

Not applicable.

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
