# Peer review of "Potential Pathogenic Impact of Cow’s Milk Consumption and Bovine Milk-Derived Exosomal MicroRNAs in Diffuse Large B-Cell Lymphoma"

_ijms, 2023, doi:10.3390/ijms24076102_

Round 1
Reviewer 1 Report
In the submitted manuscript Melnik et al. gave an extensive overview on the association between bovine milk consumption and the risk of diffuse large B-cell lymphoma, with special emphasis on milk-derived exosomal microRNAs.
This manuscript is very comprehensive and quite well written, however, there are some minor drawbacks in inconsistent writing styles:
1) Authors should uniformly italicize gene symbols throughout the text, paying special attention when they write about genes, and when about proteins, especially since miRNAs act on genes. So for instance, you cannot say "p53 transcription", but "TP53 transcription" (line 591).
2) To avoid any ambiguity, whenever that information is available from the original literature, always the full symbol of a mature miRNA must be provided! For example, there are lots of text about miR-21, but since both mature miRNAs, miR-21-5p and miR-21-3p exist (https://www.mirbase.org/cgi-bin/mirna_entry.pl?acc=hsa-mir-21), this text it too imprecise.
3) Authors should check and provide explanation in the text of all non-standard abbreviations, like IKK, HL, CI, GH, SD, etc. Also, all abbreviations presented in figures and tables must be explained in figure legends and tables' footnotes.
4) Amino acids should always be written in full non-capitalized names (e.g., lines 169, 717 and 719).
5) 95% confidence intervals should consistently be written as a range "(x–y)". Also, since lots of 95% CIs are very close to 1.0, it is questionable if those relative risks are even statistically significant. Therefore, p-values should be provided for all presented RRs.
6) When writing about mice diet, proper phrases would be, e.g., "mice fed with lactalbumin" or "lactalbumin-fed mice".
7) Line 201: It should be precisely stated if increased or decreased LAT1 expression is a poor prognostic factor. Also, it is unclear why standard deviation was use as a dispersion measure for median value of LAT1 expression?! Median is usually presented with a range.
8) Line 206: It is unclear what means "good correlation efficiency". I believe "good correlation" is sufficient.
9) Line 234: References should be put in brackets.
10) Lines 333-334. It is unclear what means "opposite prognostic factors" in sentence "ERβ1 protein expression represented opposite prognostic factors in nodal vs. extra-nodal DLBCLs".
11) Lines 772-773: Sentence "The inositol polyphosphate-5-phosphatase, 145-KD (INPP5D also known as SHIP1) is another target of miR-155" is quite incomprehensive since it is unclear what is "145-KD", and why gene and protein symbols were mixed together?!
12) Line 1185: Provide URL for PubMed.
Author Response
We thank R1 for the general appreciation of our manuscript and all constructive remarks and corrections.
We would like to respond point by point.
Ad1) We uniformly italicized gene symbols throughout the text.
Ad 2) We provided the full symbol of a mature miRNA (3p/5p), when possible and provided by the original literature.
Ad 3) We checked and provided explanation in the text of all non-standard abbreviations and explained all abbreviations in figures and tables in figure legends and tables' footnotes.
Ad 4) Amino acids names are presented as non-capitalized names.
Ad 5) We provided 95% confidence intervals consistently as a range and presented, available p-values.
Ad 6) We corrected the sentence as suggested “mice fed with lactalbumin"
Ad 7) We clarified the statement concerning LAT1 expression in DLBLC.
Ad 8) We improved this sentence accordingly.
Ad 9) We put references 110 and 111 in brackets.
Ad 10) We improved the sentence: “ For nodal lymphoma, high ERβ expression (≥25%) was associated with poorer event free survival independent of the international prognostic index with the adjusted hazard ratio (HR) of 2.49 (95% CI: 1.03-6.00, P = 0.042.”
Ad 11) Inositol polyphosphate-5-phosphatase, 145-KD is the OMIM definition of INPP5D. For a better readability we omitted “145-KD”.
Ad 12) We provided the URL for PubMed.
Reviewer 2 Report
The review focused on the link between cow´s milk consumption and the risk of diffuse large B-cell lymphoma (DLBCL) through miRNA. The authors suggested the risk of Bovine MDEs and their miR cargo as potential pathogens that should be removed from the human food chain.
My comments
-In materials and methods: the authors should indicate timeline for papers colleceted, also a flow chart is recommended. The start number of papers, how many excluded and how many included
-4.Exosomal MicroRNAs in the Pathogenesis of DLBCL: add more information regarding exosomes, their biological function and how they differ from other Extracellular vesicles
-In conclusion needs improvement. The authors highlight concerns regarding adults, recommendations regarding infancy should be extended
Author Response
We thank R2 for all supportive comments.
Ad 1) In materials and methods we presented the timeline for collected papers. As we presented a narrative review and not a systematic review, we waived a flow chart presenting inclusion and exclusion criteria.
Ad 2) We extended basic information regarding exosomes, their biological function their relation to the whole family of extracellular vesicles.
Ad 3) In the conclusion, we provided information regarding milk consumption during infancy and its potential implication in the pathogenesis of DLBCL.
Reviewer 3 Report
The authors in this review present a unique, but a rather controversial aspect on milk consumption and its correlation with developing cancers such as the DLBCL. Milk and other dairy products have been consumed since ancient times. The authors in the paper have discussed multiple milk components/factors and exosomal miRNAs linking them with the development of DLBCL. The collected information seems to be extensive and relevant. I have few major comments though:
Major comments
1) I would appreciate if the authors followed a balanced approach and discuss in the discussion section whether milk has any anticancer effects too. Is milk alone a major risk factor or it is in combination with other lifestyle choices/family history?
2) Does, boiling or pasteurization or any other treatment make milk safer?
3) Is the list on milk components that are pro-cancerous exhaustive? Please ensure that all important factors relevant to the paper are listed to support the conclusions.
Author Response
We thank R3 for all helpful comments, questions and suggestions.
Ad 1) We provided epidemiological evidence that milk consumption has protective effects for the risk of colorectal carcinoma. In colorectal carcinoma cells, miR-148a-3p is downregulated and functions as a tumor suppressor in this type of cancer. Milk-derived exosomal miR-148a-3p exerts anti-inflammatory effects in the intestine and may enhance miR-148a-3p expression.
Milk may operate in concert with other dairy products but boiling, ultraheat treatment and fermentation of milk apparently attenuates milk exosome-derived miR signaling. We added these aspects to the discussion.
We showed that commercial milk contains and transfers various components with cancerogenic activity like exosomal miRs, oncogenic viruses, estrogens, bisphenol A and others. So milk is not a single substance but a mixture of biologically active components, which may have synergistic effects in the pathogenesis of DLBCL. The interaction of commercial milk with other food components is very interesting but out of the scope of our already extensive review.
The genetic background of the milk consumer may play a critical role for milk-mediated lymphomagenesis. Individuals with lactase persistence, IGF1 polymorphisms, BRCA1 mutations etc. may experience higher oncogenic risks aggravated by milk consumption. At present, data are missing to cover this aspect.
Ad 2) Boiling and ultraheat treatment (UHT) destroys milk exosomes and their miRs in contrast to pasteurization. Thus, boiling and UHT makes milk safer because these thermal procedures eliminate milk miR signaling.
Ad 3) To our knowledge, we presented all relevant milk components that potentially promote DLBCL lymphomagenesis.
Round 2
Reviewer 2 Report
The authors addressed my comments
Reviewer 3 Report
I thank the authors for addressing my questions/comments.